# MS-DIAL 5 multimodal mass spectrometry data mining unveils lipidome complexities

Lipidomics and metabolomics communities comprise various informatics tools; however, software programs handling multimodal mass spectrometry (MS) data with structural annotations guided by the Lipidomics Standards Initiative are limited. Here, we provide MS-DIAL 5 for in-depth lipidome structural elucidation through electron-activated dissociation (EAD)-based tandem MS and determining their molecular localization through MS imaging (MSI) data using a species/tissue-specific lipidome database containing the predicted collision-cross section values. With the optimized EAD settings using 14 eV kinetic energy, the program correctly delineated lipid structures for 96.4% of authentic standards, among which 78.0% had the *sn*-, OH-, and/or C = C positions correctly assigned at concentrations exceeding 1 μM. We showcased our workflow by annotating the *sn*- and double-bond positions of eye-specific phosphatidylcholines containing very-long-chain polyunsaturated fatty acids (VLC-PUFAs), characterized as PC n-3-VLC-PUFA/FA. Using MSI data from the eye and n-3-VLC-PUFA-supplemented HeLa cells, we identified glycerol 3-phosphate acyltransferase as an enzyme candidate responsible for incorporating n-3 VLC-PUFAs into the *sn1* position of phospholipids in mammalian cells, which was confirmed using EAD-MS/MS and recombinant proteins in a cell-free system. Therefore, the MS-DIAL 5 environment, combined with optimized MS data acquisition methods, facilitates a better understanding of lipid structures and their localization, offering insights into lipid biology.

Untargeted lipidomics has emerged as a crucial biotechnology approach, enabling comprehensive lipidomic analysis of various biospecimens[1]. Tandem mass spectrometry (MS/MS) of lipids ionized by electrospray ionization (ESI), followed by collision-induced dissociation (CID)-based fragmentation, provides detailed substructure information. This allows the characterization of lipid structures at the molecular species level, characterizing lipid subclasses in addition to carbon and double bond numbers in the individual acyl chains[2]. In addition, advanced techniques such as electron-based method[3], Paterno-Buchi reaction[4], ultraviolet photodissociation[5], and ozone- or hydroxyl-radical reactions[6,7] offer deeper insights into lipid structures by annotating the *sn*-position and double bond (C = C) locations.

Moreover, spatial lipidomics, such as matrix-assisted laser desorption ionization (MALDI) coupled with MS, facilitate the determination of lipid molecule localizations[8]. Consequently, the screening, in-depth structural annotation, and spatial mapping of lipids are now feasible using state-of-the-art analytical chemistry tools. Given these advancements, the development of an informatics environment that fully leverages the potential of advanced MS techniques has become a pressing need, propelling lipid-centric biological research forward.

Despite the development of various informatics tools within the lipidomics and metabolomics communities[9], only a limited number of software programs can handle multimodal MS data with structural annotations guided by the Lipidomics Standards Initiative (LSI). In this

✉e-mail: harayama@ipmc.cnrs.fr; marita@keio.jp; htsugawa@go.tuat.ac.jp

study, we introduce MS-DIAL 5, an advanced environment that builds upon its predecessor, MS-DIAL 4[10], which supports diverse MS methodologies and has an improved user interface utility. This environment excels in multimodal MS data analysis, enabling the in-depth elucidation of lipid structure with electron-activated dissociation (EAD)[3] and facilitating spatial lipidomics through a tissue/species-specific lipid CCS database constructed using a machine learning method on datasets acquired from CID-based untargeted lipidomics studies. The MS-DIAL 5 program is validated with authentic standards in addition to well-defined biological samples such as NIST SRM 1950 plasma. Furthermore, the multimodal MS environment is showcased by the in-depth structure characterizations of eye lipidome in mice where the biosynthetic pathway for incorporating n-3 very long chain polyunsaturated fatty acids into the *sn1* position of phospholipids in mammalian cells is also elucidated.

## Results and discussion

MS-DIAL 5 was designed to handle multimodal MS data, where the main focus in this study is to develop an environment for in-depth structure elucidation using EAD (Table 1). We initially assessed the informational content of EAD-MS/MS by examining the spectra of 716 unique small molecules, which revealed that a kinetic energy (KE) of 15 eV yielded the most MS/MS information, as determined by the spectrum entropy values and molecular spectrum networks[11,12] (Fig. 1a, Supplementary Fig. 1, and Supplementary Data 1). Subsequently, we analyzed the MS/MS spectra of 65 lipids at KEs from 8 to 20 eV and selected 14 eV as the optimal KE for lipid structure analysis for three main reasons: first, the sensitivity of the product ions was higher in the 14 eV KE than in those of 8–10 eV KE, which was utilized in the previous report as the optimal parameter[3] (Fig. 1b and Supplementary Data 2). Second, while the V-shaped pattern, whose valley corresponds to the C = C position, is an important criterion for interpreting the C = C position, the MS/MS spectrum of the 14 eV KE maintains the pattern in various phospholipid subclasses (Fig. 1b–d and Supplementary Fig. 2). The potential mechanism for the increase in product ions abundance adjacent to the double bond is the stabilization of the fragment ion by the McLafferty rearrangement for the hydrogen loss (H-loss) fragment ion or allyl radical formation for the radical fragment ion[13] (Fig. 1b). As the increased peaks can be utilized as marker ions for structure elucidation, they are termed "C = C high" peaks in this study (see Supplementary Methods). Finally, the 14 eV KE condition provided unique diagnostic ions for characterizing PUFA in addition to V-shaped

patterns. We observed a significant increase in the hydrogen gain (H-gain) fragment ions in phospholipids with methylene-interrupted PUFAs containing more than three double bonds, including arachidonic acid (ARA) and docosahexaenoic acid (DHA) (Fig. 1d and Supplementary Fig. 3). An increase in the abundance of H-gain fragment ions was observed under high KE conditions (14–18 KEs), which is likely due to the McLafferty rearrangement facilitating the removal of acidic protons from the methylene moiety between double bonds. The principle of C = C position determination in EAD relies on charge-remote fragmentation (CRF), producing three ion types—H-loss, radical, and H-gain—at each carbon-carbon cleavage, complicating PUFA structural elucidation. Thus, the distinctive pattern of PUFA-specific H-gain fragments at the 14–18 eV KEs provides a key criterion for structure elucidation. Moreover, we observed benefits in the annotation of sphingolipids, with facilitated characterizations of hydroxy (OH) positions and *N*-acyl chain compositions, and in glycerolipids, enabling the discrimination of polar head isomers such as 1,2-diacylglyceryl-3-*O*-2'-(hydroxymethyl)-(*N,N,N*-trimethyl)-*β*-alanine (DGTA) and 1,2-diacylglyceryl-3-*O*-4'-(*N,N,N*-trimethyl)-homoserine (DGTS), as evidenced by their 14 eV KE spectra (Supplementary Methods, Supplementary Fig. 4, and Supplementary Fig. 5).

Through detailed investigations of the MS/MS spectra, we developed a decision tree algorithm for the 14 eV KE spectra to elucidate lipid structures that were implemented in MS-DIAL 5 (Supplementary Methods and Supplementary Note 1). Our algorithm first annotated the molecular species level of lipids, such as PC 16:0_20:4 and ceramide (Cer) 18:1;O2/16:0. If only species-level annotations, such as PC 36:4 and Cer 34:1;O2, were assigned, no further details were examined. The *sn*-, OH-, and C = C positions were independently evaluated. For the *sn*- and OH-position assessments, the candidates were ranked based on the abundance of the diagnostic product ions, with no positional assignments made if they did not match the theoretical *m/z* values. For the determination of the C = C positions within complex lipids, the presence of product ions labeled as "C = C high" peak was first confirmed. Then, as unique abundance patterns, such as the V-shape, are not the exclusive criteria, candidates were ranked based on the local correlation coefficient between the experimental and computationally generated in silico spectra related to the acyl chains. The reference spectra included heuristic H-loss, radical, and H-gain ions that reflect the V-shape and increase in PUFA-H-gain ions. To develop the MS-DIAL annotation algorithm, the peak heights of the diagnostic product ions were assessed using authentic standards of 1,2-dilinoleoyl-*sn*-glycero-

## Table 1 | Informatics software and tools in lipidomics

| Software name | MS-DIAL 5 | MS-DIAL 4 | Lipostar 2.0 | LDA2 | LipidFinder 2.0 | XCMS online | mzmine | LipidHunter2 |
|---|---|---|---|---|---|---|---|---|
| DI-MS (Full MS) | ✓ | | | | | | ✓ | |
| DI-MS (DDA) | ✓ | | | | | | ✓ | ✓ |
| DI-MS (DIA) | ✓ | | | | | | ✓ | |
| IM-MS (Full MS) | ✓ | | | | | | ✓ | |
| IM-MS (DDA) | ✓ | | | | | | ✓ | |
| IM-MS (DIA) | ✓ | | | | | | ✓ | |
| LC-MS (Full MS) | ✓ | ✓ | ✓ | ✓ | ✓ | ✓ | ✓ | |
| LC-MS (DDA) | ✓ | ✓ | ✓ | ✓ | | | ✓ | ✓ |
| LC-MS (DIA) | ✓ | ✓ | ✓ | | | | ✓ | |
| LC-IM-MS (Full MS) | ✓ | ✓ | ✓ | | | | ✓ | |
| LC-IM-MS (PASEF) | ✓ | ✓ | ✓ | | | | ✓ | |
| LC-IM-MS (diaPASEF) | ✓ | | | | | | | |
| In-depth lipid annotation using EAD | ✓ | | | | | | | |
| MSI | ✓ | | | | | | ✓ | |

The summary was created based on the publication of Ni, Z. et al. [9] Nat Methods 20, 193–204 (2023) with the actual website's information on December 11, 2023. DI: direct or flow injection (for shotgun lipidomics), *IM* ion mobility (IM), *IM-MS* direct or flow injection with IM separation, *LC* liquid chromatography, *DDA* data dependent acquisition, *DIA* data independent acquisition, *MSI* mass spectrometry imaging.

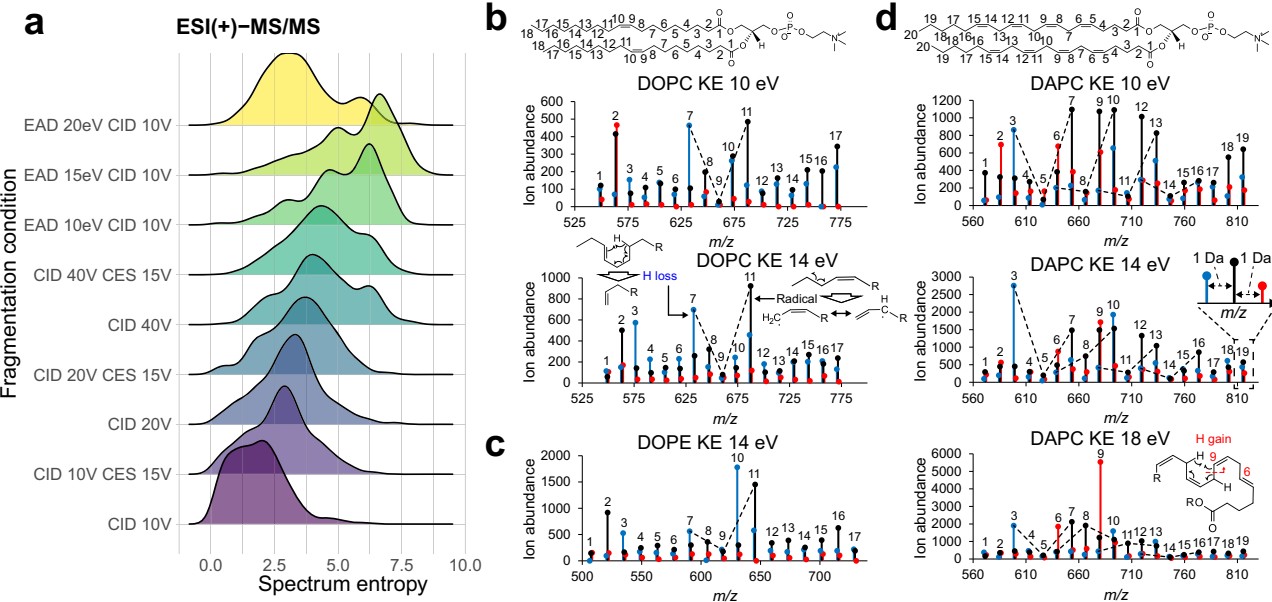

**Fig. 1 | Electron-activated dissociation (EAD)-based tandem mass spectrum facilitates efficient lipid structure elucidation. a** Spectrum entropy value distributions for 716 small molecules, with the x- and y-axes representing spectrum entropy and fragmentation conditions, respectively. **b** EAD-MS/MS spectra of 1,2-dioleoyl-*sn*-glycero-3-phosphocholine (DOPC) at kinetic energies (KE) of 10 and 14 eV, highlighting only the hydrogen (H) loss (blue), radical (black), and H-gain (red) fragment ions related to acyl chain properties. The proposed mechanism explaining the increased abundance of H-loss and radical fragments is also depicted. **c** EAD-MS/MS spectrum of 1,2-dioleoyl-*sn*-glycero-3-phosphatidyletha-nolamine (DOPE) at a 14 eV KE. **d** EAD-MS/MS spectra of 1,2-diarachidonoyl-*sn*-glycero-3-phosphocholine (DAPC) at KEs of 10, 14, and 18 eV. The mechanism behind the observed increase in H-gain fragment abundance at the delta-6 and 9 carbon positions is also illustrated. Numbers atop each fragment ion denote the carbon count remaining in a single acyl chain. Source data are provided as a Source Data file.

3-phosphocholine (DLPC) and 1-palmitoyl-2-arachidonoyl-*sn*-glycero-3-phosphocholine (PAPC) (Fig. 2a). The results suggested that a large amount of lipids need to be injected into the LC-MS system to determine *sn*- and C = C positions of complex lipids (500–1000 femtomoles for PC). Therefore, we designed an EAD spectral annotation program to rank candidates from the well-characterized lipid chemical space (see Supplementary Methods for the targeted chemical space), rather than to discover new structures. This enables one to perform a comprehensive annotation of 27 subclasses, including 15 subclasses of glycerophospholipids (GP), five sphingolipids (SP), and seven glycerolipids (GL).

We evaluated the MS-DIAL program using LC-MS/MS data from a dilution series of two types of lipid mixture (Figs. 2b, c, Supplementary Data 3, Supplementary Data 4, and Supplementary Data 5): one was the LightSPLASH mixture containing 13 lipid standards dissolved in methanol solvent; the other was a mixture of 91 lipid standards derived from UltimateSPLASH and an in-house lipid standard mixture. The latter was added to the lipid extract of a stable isotope-labeled plant (as a sample matrix). EAD spectra were acquired in data-dependent acquisition (DDA) mode. The results demonstrated that both the *sn*- and C = C-positions in glycerophospholipids (GP) and both the OH- and C = C-positions in sphingolipids (SP) were achieved at high concentrations of PC, Cer, and sphingomyelin (SM). However, determining *sn*-positions for other GP lipid subclasses proved challenging, likely because of the reduced CRF reaction associated with less charge bias in the proton and ammonium ion forms[14]. The incorporation of metal ions, such as sodium, facilitates the acquisition of simpler spectra for structural elucidation owing to a stronger charge bias in the polar head moiety. As specialized sample preparation and analytical conditions are required to efficiently acquire the metal ion forms of lipids, a comprehensive evaluation of the entire analytical system for metal-ion-based lipid structure elucidation will be conducted in future work, while we offer a platform for EAD spectral annotation of sodium

adduct forms in this study. As a result of program evaluation of lipid spectra of proton or ammonium adduct forms, wherein MS-DIAL annotates lipids at species or molecular species levels with annotations of the *sn*-position, OH-position, and C = C-position based on the quality of MS/MS spectra, the program achieved a 96.4% accuracy rate by such decision-tree-based diagnostics in the annotation process (Supplementary Data 6). Among the correctly annotated peaks, 62.1% could be characterized beyond the molecular species level, with *sn*-, OH-, and/or C = C-positional information. When evaluating the results of MS-DIAL 5 using only samples with concentrations adjusted to approximately 1 µM or more (with approximately 1000 femtomoles injected into the LC-MS system), where essential fragment ions for *sn*- and C = C position determination can be detected (Fig. 2a), *sn*-, OH-, and/or C = C positions could be obtained for 78.0% of the peaks annotated at the molecular species level (Supplementary Data 6). Misannotations of the C = C positions were often observed, especially at higher concentration ranges in triglycerides (TG), phosphatidylinositols (PI), and diglycerides (DG) with different acyl chains (Supplementary Fig. 6). Except for the case in which all acyl chains were identical, the results indicated that the ammonium adduct form of non-cationic lipids provides inefficient CRF ions, suggesting the use of metal ions, such as [M+Na]⁺, for structure elucidation. Although a misannotation of the *sn*-position was observed in PC-d5 17:0/16:1(9), the product-ion spectrum could be interpreted as a mixture of *sn1*-17:0 and *sn1*-16:1 structures, even after manual inspection. This could be due to the synthesis of byproducts from the standard compounds.

It is now recognized that cells contain various isomeric lipids with different double bond positions and/or *sn1*/*sn2* organization, which are sometimes difficult to separate chromatographically. It is therefore important to detect the presence of co-eluting lipid species when they are present and to evaluate how MS-DIAL 5 behaves when the EAD spectra are generated from co-eluting lipids. We therefore assessed the effect of metabolite co-elution on lipid structure

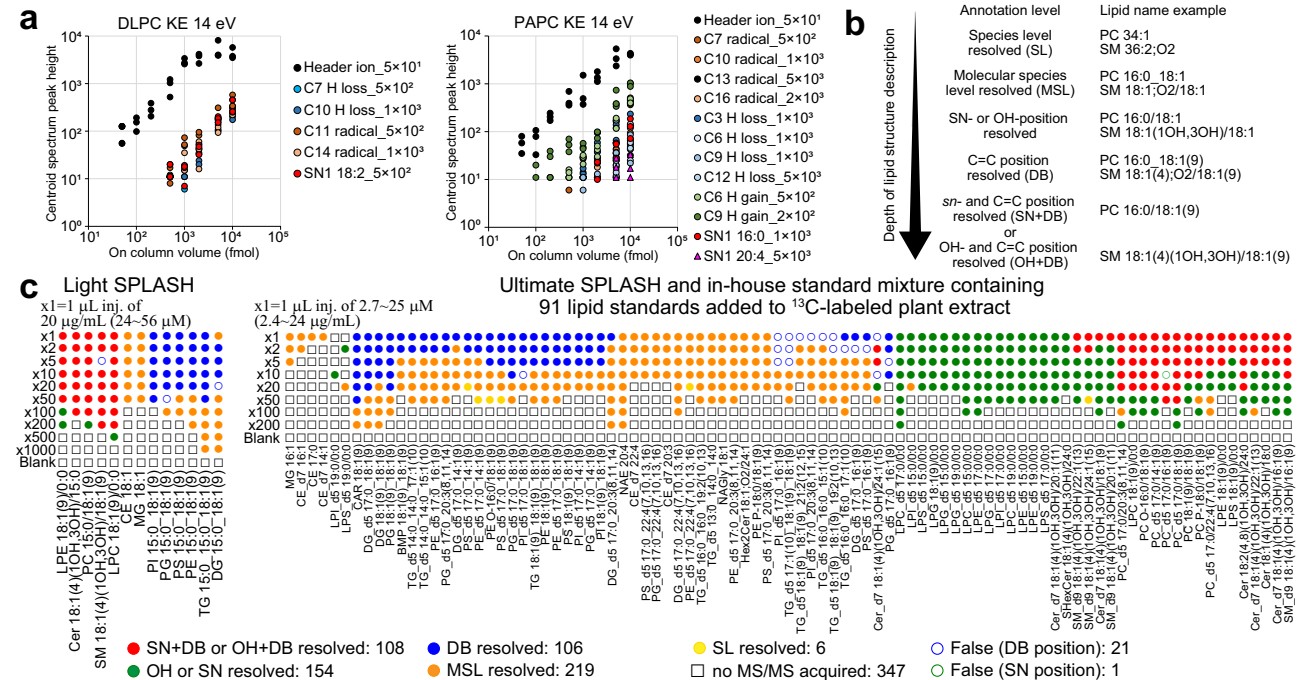

**Fig. 2 | Evaluation of dynamic ranges for in-depth lipid annotation with MS-DIAL 5. a** Dynamic range and limit of detection (LOD) required to confirm the presence of diagnostic ions for lipid structure elucidation. The x-axis represents the on-column volume of 1,2-dilinoleoyl-*sn*-glycero-3-phosphocholine (DLPC) and 1-palmitoyl-2-arachidonoyl-*sn*-phosphatidylcholine (PAPC), while the y-axis shows the peak heights of diagnostic ions from the centroided product ion spectrum obtained from the product ion scanning mode. The response values of important fragment ions were investigated; for instance, "C10 H loss_$1 \times 10^3$" and "SN1 18:2_$5 \times 10^2$" denote the LOD values of the H-loss fragment ion at the C10 position and the neutral loss (NL) of *sn1*-18:2 to characterize the *sn*-position at 1000 and 500 femtomoles (fmol) on-column volumes, respectively. Even in the authentic standard of PAPC, ions related to *sn1*-20:4 are detected due to chemical impurities or conformational changes during sample preparation. **b** Relationship between annotation level terminology and lipid description. The cases of PC and SM were described. **c** Validation of the MS-DIAL 5 environment for lipid structure

description based on EAD-MS/MS spectra quality. Dilution series were analyzed three times at each concentration. The representative annotation was determined as follows: if the same lipid name was annotated in at least two of the three replicates, that name was used as the representative annotation. If the annotation results differed across all three replicates, the lipid with the highest score was adopted as the representative. For example, "x10" indicates a dilution 10 times less concentrated than the original, denoted as "x1." For sphingolipids, green and red circles represent annotations where OH-positions and both OH- and C = C positions are resolved, respectively. For glycerophospholipids, green and red circles indicate annotations of *sn*-positions and both *sn*- and C = C-positions, respectively. Blue, orange, and yellow circles represent annotations at the C = C position resolved, molecular species, and species levels, respectively. If the MS/MS spectrum was not assigned to the precursor ion by DDA, a square shape is used. Incorrect annotations are shown as white fills with a border color indicating the source of the mis-annotation. Source data are provided as a Source Data file.

annotation using an isomeric mixture of PC 16:0/18:1(9Z), PC 18:1(9Z)/16:0, and PC 16:0/18:1(11Z) because these isomers are present in many biological samples[15–19] (Fig. 3). First, we obtained fundamental information for these lipids in terms of the spectral patterns, the precursor *m/z*-based calibration curves, and the diagnostic ion-based calibration curves (Supplementary Figs. 7a, b, and c). Then, two mixtures were prepared: a mixture of PC 16:0/18:1(9Z) and PC 18:1(9Z)/16:0, and a mixture of PC 16:0/18:1(9Z) and PC 16:0/18:1(11Z). The concentrations of the mixtures were as follows: (A) PC 16:0/18:1(9Z) and (B) PC 16:0/18:1(11Z) were adjusted to A:B = 1000 nM:100 nM, A:B = 750 nM:250 nM, A:B = 500 nM:500 nM, A:B = 250 nM:750 nM, and A:B = 100 nM:1000 nM. Mixtures for the pair of PC 16:0/18:1(9Z) and PC 18:1(9Z)/16:0 were prepared with the same ratios. The V-shapes for PC 16:0/18:1(9Z) and PC 16:0/18:1(11Z) fragment ions are generated from the *m/z* values of 606-634-661 and 634-662-689, respectively, indicating that the valley of *m/z* 634 for the annotation of PC 16:0/18:1(9Z) overlaps with the "C = C high" peak (H-loss) fragment ion of PC 16:0/18:1(11Z). Nevertheless, MS-DIAL 5 correctly characterized the higher concentration lipid isomers as the top hit candidate except for one case where the lower concentration isomer was the hit (Figs. 3a, b). Consequently, the accuracy for the annotation of the most abundant lipid molecule was 91.7% for the C = C position and 100% for the *sn*-position characterization.

Furthermore, a mixture of DLPC and PAPC was also evaluated (Supplementary Fig. 8). The structural description of PAPC was more affected by the presence of the DLPC spectrum, with proper annotation of PAPC occurring when its concentration was 5−10 times higher than that of DLPC. This effect was attributed to the identical *sn1*/*sn2* acyl chains of DLPC, which doubled the intensity of each product ion related to the acyl chains. Thus, our findings emphasize the importance of separation and lipid enrichment for in-depth structural elucidation using EAD. PAPC and DLPC can be separated in conventional LC conditions, making the above issue easy to resolve, while the isomers of PC 16:0_18:1 could not be distinguished chromatographically in our experimental condition, causing an analytical challenge (Supplementary Fig. 7a). We employed our LC-EAD-MS/MS and MS-DIAL 5 system to characterize PC 16:0_18:1 isomers in the mouse brain and NIST SRM 1950 plasma, as their abundances have been previously characterized[16,18]. The MS-DIAL 5 program characterized PC 16:0/18:1(9) as the most probable lipid isomer in both mouse brain and human plasma (Fig. 3c and Supplementary Data 7), which agrees with the previous studies[16,18]. We added a feature in MS-DIAL 5 to show other possible peak annotations, which could potentially reveal co-eluting lipids. In the brain, PC 18:1(9)/16:0 was annotated as the second potential candidate with a high confidence score, as evidenced by the detection of both diagnostic ions for the

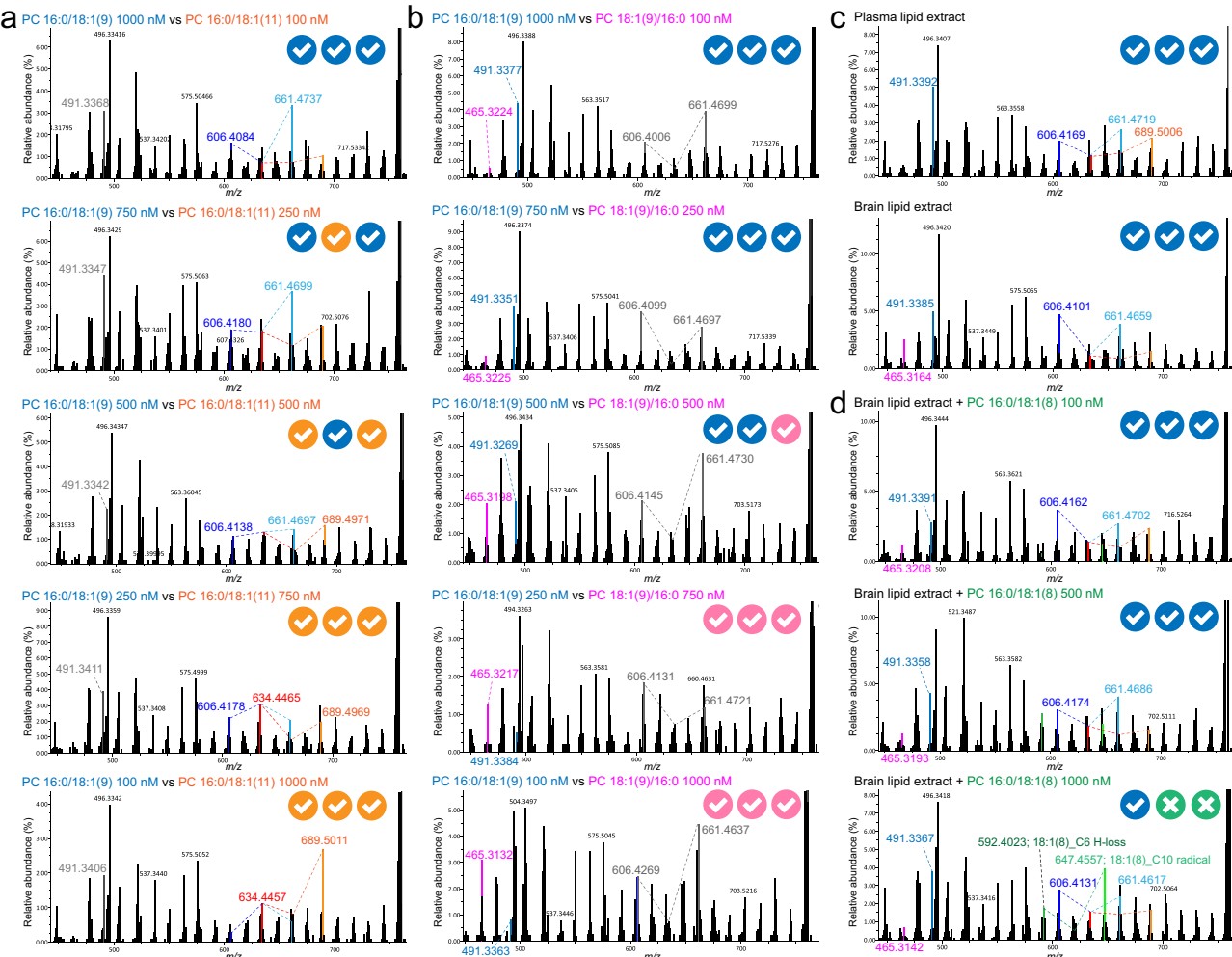

**Fig. 3 | MS-DIAL evaluation to decipher the mixed spectra of PC 16:0_18:1 lipid isomers. a** Validation of the MS-DIAL 5 environment for lipid annotation in the mixed spectra of co-eluted peaks PC 16:0/18:1(9) and PC 16:0/18:1(11). The "C = C high" peaks of the H-loss fragment ions for PC 16:0/18:1(9) and PC 16:0/18:1(11) were marked with blue and red colors, respectively. The "C = C high" peaks of the radical fragment ions for these were marked with sky-blue and orange colors, respectively. The V-shaped patterns were also described, where the valley peaks for PC 16:0/18:1(9) and PC 16:0/18:1(11) are detected at *m/z* 634.4442 and *m/z* 662.4755, respectively. The valley peak of PC 16:0/18:1(9) is completely consistent with the peak of the "C = C high" H-loss fragment ion of PC 16:0/18:1(11). When the annotations of MS-DIAL 5 were matched with the names of PC 16:0/18:1(9) or PC 16:0/18:1(11), the blue and orange check symbols were described in the upper right position of the MS/MS spectrum. The results of technical replicates 1, 2 and 3 described in Supplementary Data 7 are described by the left, middle and right

symbols, respectively. **b** Validation of the MS-DIAL 5 environment for lipid annotation in the mixed spectra of co-eluted peaks PC 16:0/18:1(9) and PC 18:1(9)/16:0. The *sn1*-specific ions for PC 16:0/18:1(9) and PC 18:1(9)/16:0 are marked with blue and pink colors respectively. When the annotations of MS-DIAL 5 were matched with the names of PC 16:0/18:1(9) or PC 18:1(9)/16:0, the blue and pink check symbols were described at the top right position of the MS/MS spectrum. The definition of the symbol replicates is the same as in Fig. 3a. **c** Validation of the MS-DIAL 5 environment for lipid annotation in mouse brain and SRM 1950 NIST human plasma lipid extracts. Color definitions are the same as in Fig. 3a and Fig. 3b. **d** Validation of the MS-DIAL 5 environment for lipid annotations in mouse brain lipid extract spiked with PC 16:0/18:1(8) at different concentrations. The 'X' symbol indicates that the MS-DIAL annotation did not match any of the PC 16:0/18:1(9), PC 16:0/18:1(11), PC 18:1(9)/16:0 and PC 18:1(11)/16:0. Source data are provided as a Source Data file.

*sn*- and C = C positions (see the detail in Supplementary Methods). PC 16:0/18:1(11) and PC 18:1(11)/16:0 were also assigned as the third and fourth highest-scoring candidates, respectively. On the other hand, PC 16:0/18:1(9) is the only candidate with a high confidence score in human plasma. Data from previous studies suggest that *sn1*-18:1 isomers, as well as isomers containing 18:1 (delta-11) are more abundant in brain than in plasma[16,18]. Thus, MS-DIAL 5 correctly identified the expected co-eluting lipids as candidates. Moreover, the abundance ratio of *sn1*-16:0 and *sn1*-18:1, along with the ratio of delta-9 and delta-11 isomers, could be estimated utilizing calibration curves made from the above mixtures of synthetic lipids (Supplementary Fig. 7d). The isomer with *sn1*-16:0 and *sn2*-18:1 (delta-9) showed high abundance in both samples, while other isomers were more abundant in brain than in plasma (Supplementary Fig. 7e),

consistently with previous studies[16,18]. To further validate our estimation of the abundances of delta-9 and delta-11 isomers, an orthogonal method using oxygen attachment dissociation (OAD)-MS/MS was employed with the same samples (Supplementary Fig. 7f, h, and i). The estimated ratio of delta-9 and delta-11 isomers gave consistent values between OAD-MS/MS and EAD-MS/MS (Supplementary Fig. 7e). Therefore, MS-DIAL 5 correctly identified abundant co-eluting lipids, with quantitative estimation being possible using synthetic standard. Further evaluation of MS-DIAL 5 was conducted using the same mouse brain lipid extract. An authentic standard, PC 16:0/18:1(8), not present in animal tissues and not included in the search space of the MS-DIAL program, was spiked to emulate contamination by unexpected molecules. MS-DIAL 5 annotated PC 16:0/18:1(9) as the top candidate in the samples spiked with 100 nM and

500 nM of PC 16:0/18:1(8) in which the endogenous PC 16:0_18:1 was adjusted to 1000 nM in LC-MS vials (Fig. 3d and Supplementary Data 7). In the sample spiked with 1000 nM, one of the three technical replicates was annotated as PC 16:0/18:1(9), while the other two were characterized as PC 16:0/18:1(11) and PC 16:0/18:1(13). From these data, we concluded that the current LC-EAD-MS/MS and MS-DIAL 5 technique can accurately annotate the most abundant lipids for co-eluted peaks, as validated for monounsaturated fatty acid-containing PC in this study. However, the assignment of double bond positions is based on changes in peak abundance patterns rather than unique fragment ions, making spectral fitting highly dependent on the availability of reference spectra. Consequently, the confidence in identifying the most abundant double bond isomer diminishes in more complex isomeric mixtures or for unsaturated lipids lacking reference spectra. The accuracy of spectral annotation can also be reduced when isomers co-elute, particularly in the case of C = C positional isomers. This highlights the necessity for further development in sample preparation and chromatography techniques to fully leverage the potential of EAD-MS/MS. Nevertheless, our current data suggests that the abundance of co-eluting lipids can be estimated if synthetic standards are available. Moreover, our results clearly indicate that EAD-MS/MS has an advantage in characterizing *sn1/sn2* positional isomers because the *sn1*-specific fragment ions are less contaminated by charge remote fragment ions related to acyl chains. This allows us to detect the presence of isomeric mixtures, such as both PC 16:0/18:1 and PC 18:1/16:0, without necessarily assigning the double bond position to the *sn*-position. The ability to annotate such mixtures is biologically significant, providing deeper insight into lipid metabolism and function.

The above experiments established the strengths of LC-EAD-MS/MS in lipid structural analysis, especially when lipid co-elution is not severe. Therefore, we demonstrated the capabilities of our in-depth lipidomic platform by characterizing phospholipids containing very long-chain polyunsaturated fatty acids (VLC-PUFAs) in the retinal tissue of mice (Fig. 4 and Supplementary Note 2)[20]. Importantly, the VLC-PUFAs are mostly contained in PC in the tissue, according to our investigation. Through an optimized solid-phase extraction procedure, we achieved an in-depth structural elucidation of 250 peaks in total and characterized 3, 20, and 10 molecules of VLC-PUFA PC at the molecular species, *sn*-position resolved, and both *sn*- and C = C-position resolved levels, respectively. While the annotation coverage was 73% lower in EAD-MS/MS than in CID-MS/MS due to lower sensitivity for product-ion peaks, MS-DIAL 5 provided in-depth structural descriptions for approximately 40% of chromatographic peaks in LC-EAD-MS/MS data: the statistical analysis was based on the results of the Bligh and Dyer evaluation (Supplementary Data 8). The interpretations of product ion spectra for highly abundant PC molecules containing PUFAs were also described (Supplementary Fig. 9). The C = C position annotations have been supported by an orthogonal approach using oxygen attachment dissociation (OAD)-MS/MS[7]. The most abundant peak in the VLC-PUFA PC fraction was characterized as PC 34:6(16,19,22,25,28,31)/22:6(4,7,10,13,16,19). Moreover, our results indicated that all the top hit candidates with *sn*- and C = C-positional information contained n-3-VLC-PUFA at the *sn1*-position, while our method does not exclude the possibility of other isomers being present. Previous indirect evidence for VLC-PUFA PC structures using phospholipase enzymes and gas chromatography-MS predicted the major forms of *sn*- and C = C-positions to be *sn1* and methylene-interrupted n-3 fatty acid, respectively[21]. In contrast, the present study directly identifies structures in their native form, and our result strongly suggests that the VLC-PUFAs are enriched at the *sn1*-position of PC in the retinal tissue, while it remains unknown how the *sn1* position-specific incorporation is achieved.

To investigate the localization of VLC-PUFAs, we reanalyzed a public MALDI-MSI dataset of eye tissues from C57B6/J and acyl-coenzyme A synthetase (ACSL) 6 knockout mice, where ACSL6 is known to have substrate specificity for DHA, by utilizing an eye-specific lipid database containing predicted CCS values generated by a machine learning model and publicly available CID-based untargeted lipidomic data (Figs. 5a, b, and Supplementary Data 9)[10,22]. The analysis revealed that the VLC-PUFA PC containing stearic acid, annotated as PC 34:6(16,19,22,25,28,31)/18:0, was not significantly reduced in ACSL6 KO mice, which was consistent across both MSI and untargeted lipidomics data (Fig. 5c and Supplementary Data 10). These results suggested that n-3-VLC-PUFA does not undergo the same enzymatic substrate recognition as DHA.

To further test whether the *sn1* specificity of VLC-PUFA incorporation requires retina-specific metabolic pathways or not, we fed HeLa cells with n-3-VLC-PUFA and evaluated its incorporation. As a result, n-3-VLC-PUFA incorporation into lysophosphatidic acid (LPA), PC, diacylglycerol (DG), triacylglycerol (TG), and cholesteryl ester (CE) was confirmed in HeLa cells when n-3-VLC-PUFA was supplemented to the culture medium (Fig. 6a, Supplementary Note 3 and Supplementary Note 4). The n-3-VLC-PUFA was incorporated into the *sn1* position of PC in HeLa cells, as confirmed by EAD-MS/MS spectrum analysis (Supplementary Fig. 10). These findings indicate that n-3-VLC-PUFA is not incorporated into phospholipids by retina-specific enzymes. Instead, the structure of n-3-VLC-PUFA, with more than 32 carbons, resembles that of saturated fatty acids, such as palmitic acid, from the carboxylic acid terminus to the first C = C-position. We hypothesized that VLC-PUFAs are recognized by glycerol 3-phosphate acyltransferase (GPAT), which prefers saturated fatty acids as substrates and incorporates an acyl chain at the *sn1*-position[23]. A cell-free system assay[24] showed that VLC-PUFAs are converted to VLC-PUFA-LPAs by recombinant GPAT1 (GPAT1$^{WT}$), which is highly expressed in the mouse retina (Supplementary Fig. 11), where acyl-coenzyme A (CoA) is synthesized by an acyl CoA synthetase in a cell-free system using wheat extract (Figs. 6b, c, Supplementary Fig. 12, Supplementary Data 11, Supplementary Note 5, and Supplementary Note 6). In contrast, the LPA molecule was not synthesized by the inactive mutant (GPAT1$^{H230A}$), the DHA-LPA molecule was not detected, and the DHA-CoA metabolite was synthesized at high levels in the cell-free system compared to those of the VLC-PUFA-CoA molecule. While the enzymatic pathways for VLC-PUFA incorporation into lipids were previously unknown, our results suggest enzymes with preferences for saturated substrates, such as GPAT1, mobilize VLC-PUFAs. The lack of double bonds near the carboxyl group makes VLC-PUFAs appear more 'saturated-like,' potentially contributing to their dominant localization at *sn1*-positions by GPATs.

The biosynthesis and function of VLC-PUFAs have been studied for very long-chain fatty acid elongase 4 (ELOVL4); however, the mechanism of VLC-PUFA incorporation into the *sn1* position of PC has not been elucidated so far. Given that the n-3-VLC-PUFA was mostly incorporated into the *sn1* position of PC in HeLa cells, the *sn*-position of VLC-PUFA in LPA can be considered as LPA *sn1*-VLC-PUFA. Thus, our proposed pathway for the most abundant VLC-PUFA PC molecule, i.e., PC 34:6(16,19,22,25,28,31)/22:6(4,7,10,13,16,19) involves ELOVL4 for VLC-PUFA synthesis, acyl CoA synthetases (ACSLs) for VLC-PUFA-CoA synthesis, GPATs for VLC-PUFA LPA synthesis, and 1-acyl-sn-glycerol-3-phosphate acyltransferase 3 (AGPAT3) for the DHA incorporation into the *sn2* position of PC. Given that the substrate recognition sites of GPAT1, ELOVL4, and ACSLs are in the cytoplasm, these three enzymes can feasibly share the substrates within the cytoplasm. The VLC-PUFA produced by ELOVL4 would be converted into VLC-PUFA CoA by an ACSL other than ACSL6. While we cannot rule out the possibility that retina-specific incorporation pathways for VLC-PUFAs exist, our platform identified that *sn1* specificity is also achieved in cultured cells, leading to the discovery of GPATs as enzymes sufficient for VLC-PUFA incorporation into glycerophospholipids. Further investigations for biochemistry related to ACSL1 - 6, ELOVL4, and the isozymes of GPAT

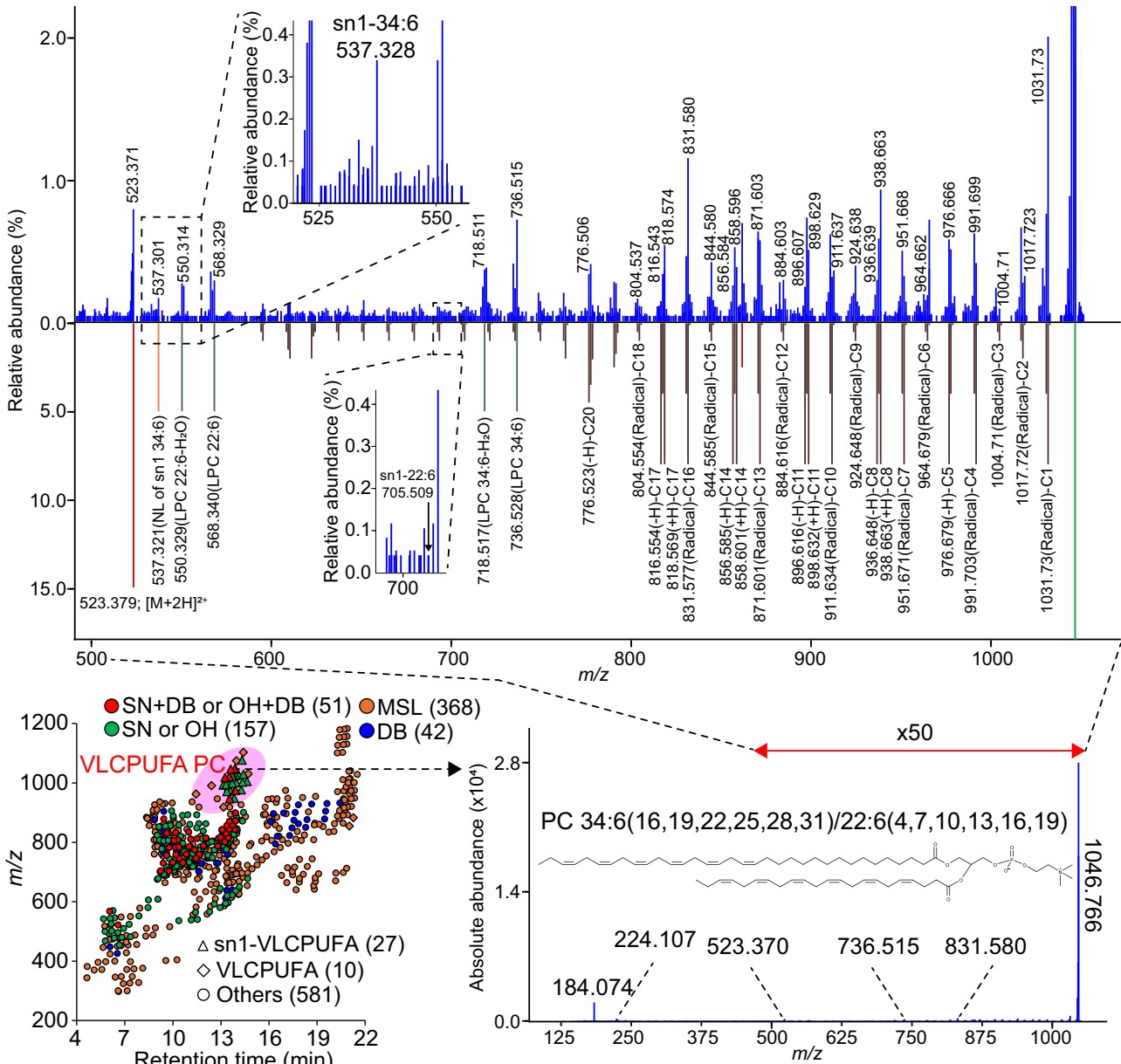

**Fig. 4 | Structural elucidations of very long-chain PUFA-containing phosphatidylcholine (PC).** The result of in-depth lipidome profiling is shown by the scatter plot of retention time- and *m/z* axis. The annotation results of molecular species (MSL), double-bond (DB) resolved, *sn*- or OH-positions (SN or OH) resolved, and both *sn*- and DB- or both OH- and DB-position (SN + DB or OH + DB) resolved levels are described by the same color charts used in Fig. 2b. The *sn1*-position determined or uncharacterized for VLC-PUFA is described by triangle and diamond symbols, respectively. The bottom-right panel describes the experimental spectrum of the lipid ion annotated as PC 34:6(16,19,22,25,28,31)/ 22:6(4,7,10,13,16,19), where the *E/Z* isomer definition in acyl chains is unsupported, yet a representative form is shown. The top panel displays a 50-fold zoomed experimental spectrum and a 10-fold zoomed in silico spectrum of the assigned lipid in the upper and lower panels, respectively. Brown, green, orange, and red spectral peaks represent ions related to homolytic cleavages in acyl chains, lyso PC substructures, neutral loss of *sn1*-34:6 moiety, and precursor- or polar head-specific fragments, respectively. Source data are provided as a Source Data file.

in the VLC-PUFA PC biosynthesis are needed. We believe that the discovery that VLC-PUFAs are incorporated into PC via GPAT is challenging to validate because the knockout or knockdown of all GPATs in cells or organisms is lethal. Nevertheless, once the enzymes responsible for VLC-PUFA uptake in the lipid remodeling pathway are identified, the contribution and role of de novo and remodeling pathways in VLC-PUFA PC biosynthesis in the eye can be elucidated.

Lipidomics has become an essential tool in systems biology and is widely used in basic research and clinical studies. Despite the structural complexity of lipids, the multimodal mass spectrometry techniques allow one to illuminate the diversity of lipids by using various methods, including untargeted analysis, in-depth structural elucidation through fragmentation methods, and spatial lipidomics. Concurrently, with the evolution of measurement techniques, the development of informatics technology has become indispensable. Since 2015, we have been developing MS-DIAL[25], enhancing not only the algorithm based on feedback from the metabolomics and lipidomics community but also creating a user-friendly interface for beginners and mature scientists. In this study, we demonstrated that MS-DIAL enables straightforward knowledge extraction from EAD spectra, spatial lipidomics, and publicly available untargeted lipidomic data, leading to insights into lipid biology. Our group aims to continue

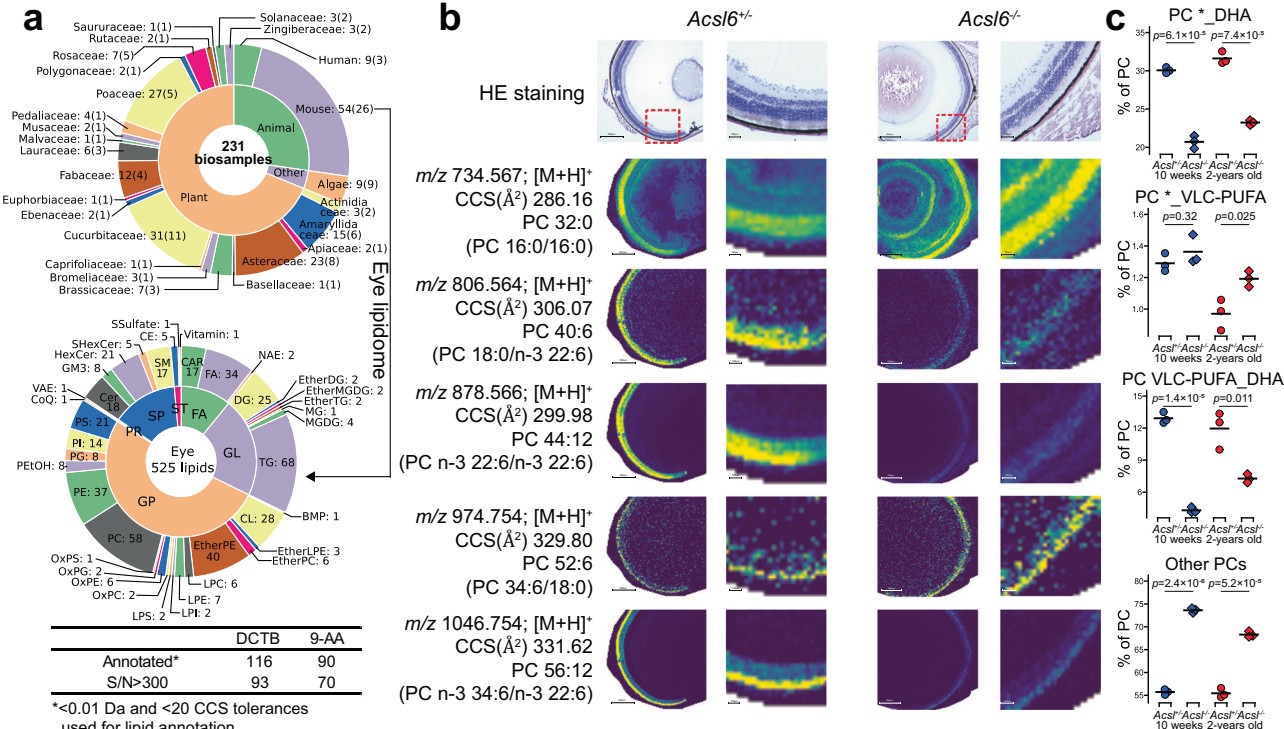

**Fig. 5 | Reanalysis of publicly available spatial- and untargeted lipidomics data for mouse eye tissues. a** A sunburst plot summarizing species/tissue-specific lipid database statistics containing collision-cross section (CCS) values. An eye-lipidome table with *m/z* and CCS values for 525 unique lipids was used to annotate lipids in MSI data analysis. A summary table of peak annotations in the analyzed MSI data is also provided. The abbreviations of FA, GL, GP, PR, SP, and ST mean fatty acyls, glycerolipids, glycerophospholipids, prenol lipids, sphingolipids, and sterol lipids, respectively. The colors in the sunburst plots were automatically generated and do not have any specific meaning. **b** Hematoxylin and eosin (HE) staining and MSI data in eye tissues from *Acsl6*+/- and *Acsl6*-/- mice. Ion distributions for five lipid molecules are shown. The reference *m/z* and CCS values for each lipid molecule are listed, with

EAD-MS/MS-based annotations for each precursor *m/z* value in parentheses (*n* = 2 biologically independent samples available at the public repository). **c** Reanalysis of publicly available untargeted lipidomics data examining eye tissues from *Acsl6*+/- and *Acsl6*-/- mice at 10 weeks and 2 years of age (*n* = 3 biologically independent samples). Here, "22:6" denotes DHA, while "28:6," "30:5," "32:4," "32:5," "32:6," "34:4," "34:5," "34:6," "36:6," and "38:6" are defined as VLC-PUFAs. An asterisk indicates acyl chains other than DHA and VLC-PUFA, with the sum of lipid molecules labeled '*_DHA' or '*_VLC-PUFA'. 'Other PCs' refers to the total abundance of PC molecules not containing DHA or VLC-PUFA). Source data are provided as a Source Data file.

contributing to data standardization in various omics sciences and developing data-driven knowledge generation platforms, facilitating machine learning and natural language processing research utilizing omics data.

## Methods

### MS-DIAL development environment
The MS-DIAL development environment was redesigned to enhance scalability, sustainability, and community contribution to the program package. The software was programmed in C#. The underlying algorithm was constructed in .NET Standard 2.0 framework, while the user interface is developed using the Windows Presentation Foundation (WPF) and follows the Model-View-ViewModel (MVVM) architecture. As a result, MS-DIAL functions as an operating system (OS)-independent command line tool, although its graphical user interface is available only for Windows OS. The source code is publicly accessible from the GitHub repository (https://github.com/systemsomicslab/MsdialWorkbench). The mzML[26] and netCDF parsers are deposited as a NuGet-package; however, owing to licensing restrictions, raw data providers for proprietary formats, such as WIFF, RAW, and .D remain private. Nonetheless, MS-DIAL supports the direct import of vendor formats, such as SCIEX, Bruker, ThermoFisher, Shimadzu, Waters, Agilent, and Kanomax. MS-DIAL employs the vendor's API to convert the original profile spectra into centroid spectra. The raw data parser is designed to retrieve centroid spectra. Consequently, users importing vendor-specific raw data into MS-DIAL should select "centroid" as the

data acquisition type for both MS1 and MS2. Conversely, when working with mzML- and Analysis Base File (ABF) format data containing profile mode spectra, users should select "profile."

### MS-DIAL 5 major functional updates when compared to MS-DIAL 4
The latest update of MS-DIAL includes significant enhancements to both the algorithmic (backend) and user interface (front-end) components. The software supports data processing for direct infusion mass spectrometry (DI-MS), direct infusion coupled with ion mobility separation (IM-MS), and mass spectrometry imaging (MSI) data. Additionally, it accommodates the data-independent acquisition of IM-MS/MS and LC-IM-MS/MS, such as the diaPASEF (parallel accumulation-serial fragmentation) used in Bruker instruments[27]. For metabolite annotation, the program includes an annotation pipeline for lipidomics employing oxygen attachment dissociation (OAD)[7] and electron-activated dissociation (EAD)[3], with the algorithm for EAD-based lipid structure elucidation being validated in this study (refer to subsequent sections).

MS-DIAL 5 is designed to cater to a wide range of users, ranging from novices to experienced analysts. Here, we introduce the front-end features tailored for seasoned analysts. Unlike MS-DIAL 4, which allows only one library file for metabolite annotation, version 5 has no such limitation. Users can load multiple MSP files containing spectra from standards generated in their laboratories, public and commercial libraries such as MassBank[28] or NIST, and in silico spectra produced

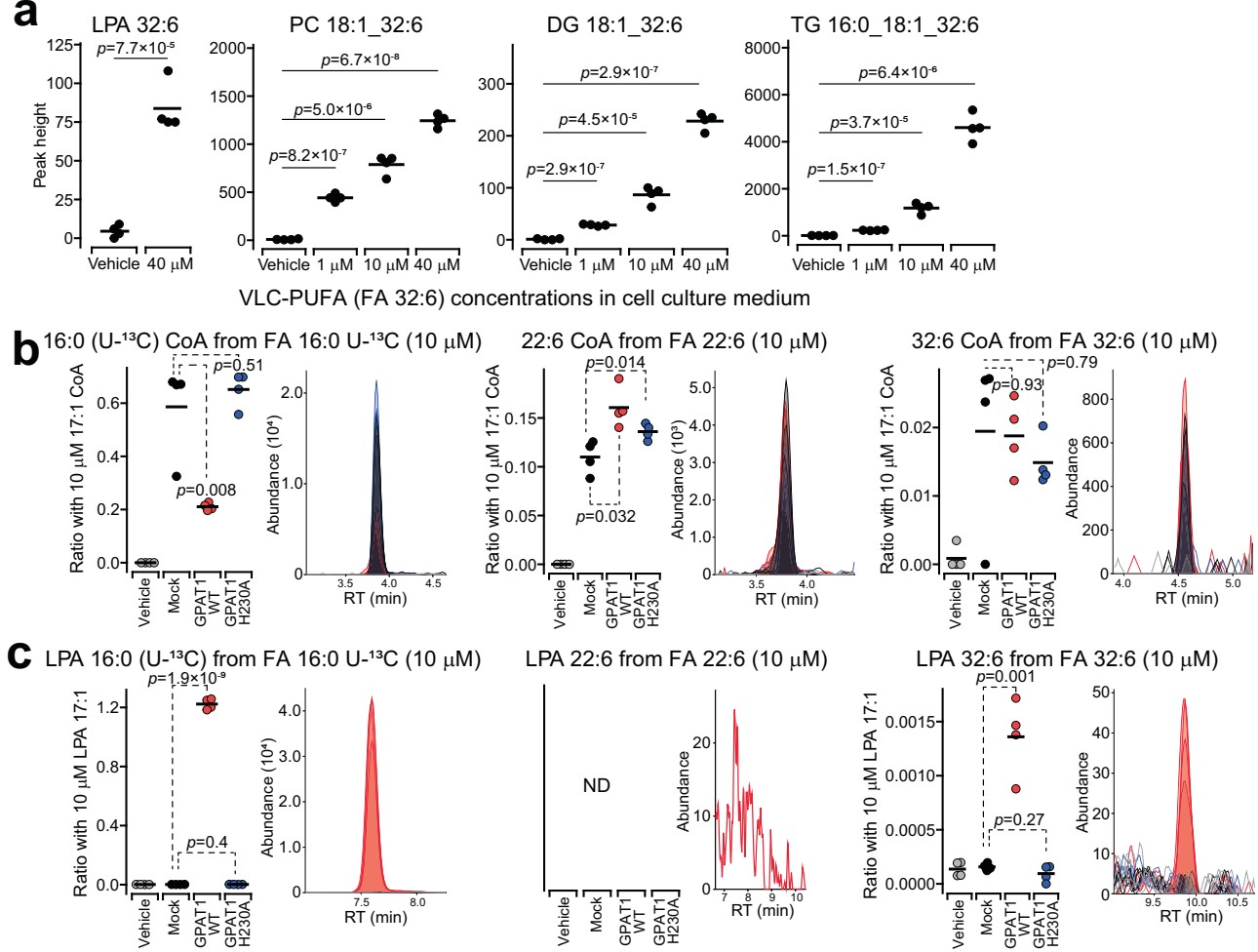

**Fig. 6 | Elucidation of VLC-PUFA PC metabolic pathway. a** HeLa cell lipid profiling with VLC-PUFA (FA n-3-32:6) supplementation (*n* = 4 biologically independent samples). The peak heights of PC 18:1_32:6, DG 18:1_32:6, LPA 32:6, and TG 18:1_18:1_32:6 at final concentrations of 1, 10, or 40 μM of FA n-3-32:6 supplementation are depicted. While LPA was analyzed by a derivatization method using trimethylsilyl-diazomethane, which converts LPA to bis-methyl LPA (BisMeLPA), other molecules were analyzed using conventional untargeted lipidomics methods. Acyl CoA (**b**) and LPA (**c**) profiling for the glycerol 3-phosphate acyltransferase 1 (GPAT1) recombinant enzyme assay (*n* = 4 independently prepared samples). The acyl CoAs and LPAs were analyzed with the vehicle, mock (native plasmid vector),

active GPAT1$^{WT}$, and the inactive GPAT1 mutant (GPAT1$^{H230A}$), supplied with glycerol 3-phosphate and coenzyme in addition to $^{13}$C-uniformly labeled palmitic acid (FA 16:0 U-$^{13}$C), docosahexaenoic acid (DHA, FA 22:6), or FA n-3-32:6, in the cell-free system enzymatic reaction. The fatty acid was supplied at a final concentration of 10 μM, and the same amount of 17:1 CoA (Fig. 6b) and LPA 17:1 (Fig. 6c) was supplied as the internal standards. The putative ratio between the converted product and the internal standard was used for the y-axis value of dot plots. The mean values were also described in dot plots. Significances were adjusted by false discovery rate in the student's t-test (two-sided). Source data are provided as a Source Data file.

using tools such as CFM-ID[29]. These files can be searched for various parameters and assigned priority levels, thereby enhancing the accuracy of the metabolite annotations and reducing false annotations in untargeted analyses.

Additionally, a companion application, "rawdataviewer.exe," is bundled with the MS-DIAL software packages. This utility provides a platform similar to SeeMS from the ProteoWizard community[26], allowing users to view raw data and adjust peak-picking parameters. For instance, the default threshold for the 'minimum peak amplitude,' which is a critical peak-picking parameter, was set to 1000. However, this value may not be optimal for instruments such as Orbitrap MS and FT-ICR-MS, where suitable thresholds often range from 10,000 to 100,000. The optimal thresholds also vary depending on the number of biospecimens and the sample matrix background. Thus, assessing the relationship between the number of detected peaks and the "minimum peak amplitude" threshold is crucial in untargeted analyses. Furthermore, exploring the relationship between the peak height/area and the signal-to-noise ratio is vital. The "rawdataviewer.exe"

application facilitates access to this information. For detailed descriptions of other utilities, please visit https://systemsomicslab. github.io/msdial5tutorial/.

## Investigating the kinetic energies to acquire information-rich EAD spectra of small molecules

MS/MS spectral records for 953 standard compounds, including the IROA large-scale metabolite library (https://www.iroatech.com/large-scale-metabolite-library-of-standards/) and an in-house natural product compound library which have been reported previously[30] were investigated. Chemical ontologies were defined using ClassyFire program[31]. These records were acquired under various fragmentation conditions (Supplementary Data 1). For EAD data acquisition, a liquid chromatography (LC) system consisting of a SCIEX Exion LC system and mass spectrometry (MS) detection of molecules were performed using quadrupole/time-of-flight MS (ZenoTOF 7600; SCIEX, Framingham, MA, USA). An InertSustainSwift C18 column (30 mm × 2.1 mm; 3 μm) from GL Sciences, Japan, was used, maintained at 40 °C with a

flow rate of 0.2 mL/min. The mobile phases were composed of (A) water ($H_2O$) with 0.1% formic acid and (B) acetonitrile (ACN) with 0.1% formic acid, utilizing an isocratic mode of 80% B. Sample temperature was kept at 4 °C. A targeted MS/MS scanning mode, referred to as "MRM HR" by SCIEX, was used. Target precursor *m/z* values were calculated for the protonated form of each molecule. The TOF mass range was set from a starting mass of 50 to an end mass equal to the precursor *m/z*. Nine fragmentation conditions were explored: three in collision-induced dissociation (CID) mode with collision energy (CE) set at 10, 20, and 40 volts (V) with no CE spread, three in CID mode with the same CEs but with a CE spread of 15 V, and three in EAD mode with CE set at 10 V and electron kinetic energy (KE) at 10, 15, and 20 electron volts (eV), all with a CE spread of 0. The accumulation times were 50 and 100 ms for the CID and EAD modes, respectively. In the EAD mode, the electron beam current and electron transfer coefficient (ETC)% were set to 7000 nA and 100%, respectively. Other parameters included ion source gas 1 at 50; ion source gas 2 at 50; curtain gas at 35; CAD gas at 7; temperature at 450 °C; spray voltage at 5500 V; and declustering potential at 80 V. Mass calibration was conducted automatically using the SCIEX Calibration Delivery System (CDS). Finally, the spectra of 716 chemicals were successfully captured and accessible at the RIKEN DROPMet.

The acquired spectrum was processed as follows: MS/MS spectral peaks accumulated across the elution fraction from the left to the right edge of the molecule's peak. A bin size of 0.05 Da was set for accumulation, and the average intensity for each bin was used as the representative value. Subsequently, the spectrum from a retention time of approximately 30 s, where no peaks were observed, was accumulated using the same method and subtracted from the compound spectrum. If the subtracted intensity was below zero, it was replaced with zero. The EAD-MS/MS spectra data were assessed using the spectrum-entropy calculation method[11]. Density plots for each fragmentation condition were visualized using ggplot2 and related packages in the R programming environment. In addition, the results of molecular spectral networking from CID at 40 V with a collision energy spread (CES) of 15 and EAD at 15 eV are detailed. The MS/MS spectral similarity among the compounds was calculated using a modified dot product score as previously reported[12]. The parameters for this calculation were as follows: relative abundance cutoff of 0.1%, absolute abundance cutoff of 50, product ion mass tolerance of 0.05, mass binning value of 1.0, intensity scale factor of 0.5, and a maximum scale value of 100. The source code for this process can be found in the MsScanMatching.cs file in the MsdialWorkbench repository.

### Investigating the kinetic energies to obtain information-rich EAD spectra for lipid structure elucidation

MS/MS spectral records were obtained for 34 glycerophospholipid (GP), 18 molecules of sphingolipid (SL), 9 molecules of glycerolipid (GL), 1 molecule of sterol lipid (SL), and 3 molecules of fatty acyls (FA) molecules (Supplementary Data 2). The mobile phases included (A) a mixture of ACN, methanol (MeOH), and $H_2O$ in a 1:1:3 (v/v/v) ratio with 5 mM ammonium acetate and 10 nM ethylenediaminetetraacetic acid (EDTA) and (B) a 1:9 (v/v) mixture of ACN and isopropanol (IPA) with 5 mM ammonium acetate and 10 nM EDTA. The other LC conditions remained unchanged. Nine fragmentation conditions were explored during the targeted MS/MS scanning mode, including one CID setting (45 eV) and various EAD settings with different kinetic energies (8, 10, 12, 14, 16, 18, and 20 eV KE). A constant CE spread of 0, collision energy of 12 for EAD, and a time-of-flight (TOF) start mass of 170 were applied. The accumulation time was 100 ms for both CID and EAD modes. The ion source temperature was maintained at 275 °C, while other parameters were consistent with those described above.

The charge remote fragment (CRF) ion pattern of the lipid molecules was elucidated using the EAD-MS/MS spectra of the three lipid metabolites, as shown in Fig. 1 of the main text. CRF ions arise

from the homolytic cleavage of chemical bonds, producing three types of fragment ions: hydrogen loss (H-loss), radical, and hydrogen gain (H-gain) derived from each carbon-carbon bond cleavage. As demonstrated in a previous report[3], a KE of 10 eV yields a distinct V-shaped pattern in the product ion intensities around the double-bond position. For instance, the radical ion abundance at the "C9" position from the C9-C10 bond cleavage in dioleoyl PC (DOPC) is reduced, while the ion abundances at the "C11" and "C7" positions from the C11-C12 and C7-C8 bond cleavages, respectively, are increased (Fig. 1b). The term "C = C low peak" is used in this paper to denote the low-intensity peak. The amplified abundance of radical ions at the "C11" position is interpreted as stabilization of the fragment ion by resonance structure formation. Additionally, the significant increase in the C7 fragment ion intensity suggests that the hydrogen of C6 is more readily transferred by the electron pair of the double bond between C9 and C10 because of the McLafferty rearrangement. Such markedly increased fragment ions are referred to as "C = C high peak" in this study. Fragment ion abundances in EAD-MS/MS became more pronounced at KE 14 eV than at KE 10 eV, while preserving the V-shaped pattern. Comparable patterns were observed in dioleoyl PE (DOPE) and diarachidonoyl PC (DAPC) (Fig. 1c, d). However, when complex lipids include PUFAs, such as DAPC, pinpointing the valley in the V-shaped product ion pattern is challenging owing to increased spectral complexity.

In the EAD-MS/MS spectrum derived from the KE 18 eV condition, unique fragmentation patterns emerged that were not present at KE 10 eV. Specifically, in the KE 18 eV MS/MS spectrum of DAPC, there was a marked increase in the abundance of the hydrogen gain (H-gain) fragment ion at the C9 position (Fig. 1d). This phenomenon of increased H-gain fragment ion is also notable in structures like linolenic acid, arachidonic acid, and docosahexaenoic acid, which possess multiple methylene-interrupted C = C bonds, confirmed by the *Z* isomer. The hydrogen atom on the methylene group situated between the double bonds is more acidic than the typical C-H bond, which is known to be an acidic proton. The observed intensification of the H-gain fragment ion is attributed to the effective transfer of this acidic proton to the electron of the double bond, facilitated by McLafferty rearrangement, as demonstrated in the acyl chain of arachidonic acid. Thus, the pronounced increase in the H-gain fragment ion abundance under KE 18 eV conditions serves as a diagnostic marker to differentiate C = C-positional isomers, such as n-3, n-6, and n-9 fatty acid chains. This increase in H-gain fragment ion is described as "C = C PUFA high" in this study. Notably, an increase in the abundance of H gain fragment ions was detected in the EAD-MS/MS spectrum at KE 14 eV. Therefore, the KE 14 eV setting provides (1) enhanced sensitivity of the product ions, (2) a characteristic V-shaped pattern in lipids with C = C bonds, and (3) a distinct H-gain ion behavior from PUFAs with more than three C = C bonds.

### Overview of MS-DIAL lipid annotation for EAD-MS/MS

This program employs a decision-tree-based method to annotate each lipid subclass. Given that a collision energy (CE) of approximately 10 V is commonly applied to ion transfer within mass spectrometers, including in EAD mode, the product ion spectrum from EAD represents a composite of CID-based fragmentation and charge-remote fragmentation (CRF) effects on lipids. Consequently, the program initiates the annotation process by searching for a diagnostic ion or neutral loss characteristic of the lipid subclass, which aids in determining the acyl chain attributes. The feasibility of identifying *sn*-positions and hydroxy (OH) groups varies with the lipid subclass and the type of adduct formed. For instance, the fragment ion from the homolytic cleavage at the C1-C2 bond of the glycerol backbone, indicative of the *sn*-position in glycerolipids (GLs) and glycerophospholipids (GPs), is readily detectable in the protonated form of PCs. However, the neutral loss fragment of the *sn1*-acyl chain from the protonated or ammonium adducts in other GLs and GPs may be less distinguishable from the acyl

chain CRF- or noise ions. Because the cationic moiety in lipids tends to stabilize in the sodium adduct form, the EAD-MS/MS spectra for sodium adducts tend to be less convoluted than those for the $[M + H]^+$ and $[M + NH_4]^+$ forms. In this study, the annotation pipeline in MS-DIAL was assessed for $[M + H]^+$ and $[M + NH_4]^+$ adducts, which are predominant in conventional untargeted lipidomics. On the contrary, the annotation for $[M+Na]^+$ adduct is available in the current MS-DIAL 5 platform, with its validation for sodium adducts in combination with the optimization of sample preparation and analytical chemistry to be reported elsewhere. Furthermore, the elucidation of the OH position, which is vital for understanding sphingolipid metabolism, was enhanced by EAD in tandem with the MS-DIAL computational framework, given the distinct visibility of OH-position-related fragment ions.

The procedure for determining the position of the carbon-carbon double bonds (C = C) is as follows. Initially, the search is conducted for the presence of "C = C high" peaks in the product ion spectrum. Typically, for each C = C location, two such peaks are expected, with an additional "C = C PUFA high" peak observed for PUFA acyl chains containing more than three C = C bonds. If any of the expected "C = C high" peaks are absent from the product ion spectrum, the candidate molecule is eliminated from consideration. If more than five "C = C high" peaks were predicted, indicating the presence of three or more double bonds within the acyl chain, the algorithm was designed to allow annotation of double bond positions even if one of the "C = C high" peaks was missing. For example, if there were three double bonds, six "C = C high" peaks were expected (or seven if the acyl chain contained a methylene-interrupted C = C sequence, such as linolenic acid, producing "C = C PUFA high" peak). The program would attempt to determine the double bond positions even if one of these six peaks was not detected; that is, if two or more of the six peaks were missing, the program would not proceed with double bond position annotation. Subsequently, the correlation between the experimental and *in-silico* spectra of the lipid molecules was calculated, focusing on the CRF ions of the acyl chains. The *in-silico* spectrum generation involved computing *m/z* values for hydrogen loss (H-loss), radicals, and hydrogen gain (H-gain) fragment ions for each homolytic cleavage along the acyl chain. The intensity ratios for the H-loss, radical, and H-gain fragment ions were set to 0.5, 1, and 0.05, respectively, for saturated fatty acyl chains. These ratios are adjusted to 0.25, 0.5, and 0.05 for a "C = C low" peak, and to 2.0, 4.0, and 0.05 to reflect an H-loss peak increase, and 4.0, 2.0, and 0.05 to indicate a radical peak increase in a "C = C high" peak. The factor of the H-gain fragment was changed to 4.0 for the "C = C PUFA high" peak. A "reverse dot product similarity score", utilizing the *in-silico* spectrum as the library template, was employed as a measure of correlation. The "matched fragment peaks percentage" between the experimental spectra and in-silico spectra was also used for ranking the candidates. In addition, the average peak intensity is calculated for the peaks annotated as "C = C high" and "C = C low" peaks. If the average intensity of the "C = C high" peaks is more than 1.5 times greater than that of the "C = C low" peaks, a bonus score of 0.5 is added: this is defined as the "V-shaped bonus score". Furthermore, for PUFAs containing three or more double bonds, if the intensity of the peak annotated as "C = C PUFA high" is more than three times the average intensity of the other H-gain fragment ions, an additional bonus score of 0.5 is added: this is defined as the "PUFA H-gain bonus score". In summary, MS-DIAL currently calculates the EAD-based annotation score named as "Matched Peaks Percentage" described in Supplementary Data 7 and uses it to rank annotation results based on EAD-MS/MS as follows.

$$\begin{aligned}\text{"EAD-based annotation" score} = &\text{"Class" score} + \text{"Chain" score} \\ &+ \text{"Position" score} + \text{"Double Bond" score}\end{aligned} \quad (1)$$

In the above equation, the class, chain and position scores are one if the diagnostic ions are detected and zero if they are not. In addition,

the double bond score is calculated as follows.

$$\begin{aligned}\text{"Double Bond" score} = &\text{"Reverse dot product similarity" score} \\ &+ \text{"Matched Fragment Peaks Percentage" score} \\ &+ \text{"V - Shape bonus" score} + \text{"PUFA H - gain bonus" score}\end{aligned}$$
$$(2)$$

The maximum of "Double Bond" score is three and the maximum of "EAD-based annotation" score is six. Based on the results validated in Fig. 3, MS-DIAL tends to give incorrect results when this score is less than 4.8. Thus, in this study, the EAD-based annotation score, referred to as "Matched peaks percentage" in Supplementary Data 7, with more than 4.8 is recognized as a high confidence score for the annotation of both *sn*- and C = C positional isomers. The candidates were then ranked according to their "EAD-based annotation" scores, with the highest scoring candidate designated as the representative isomer in the EAD-MS/MS spectrum.

The methodology for defining noise ions ("barcode ions") in MS-DIAL 5 is described. The profile spectrum was centroided by the SCIEX application programming interface (API). In the centroided product-ion spectra obtained from the SCIEX ZenoTOF 7600 instrument, the product-ion peak with an intensity less than 11 was defined as noise ion (barcode ion) because the peaks having 11 ion intensity were observed randomly in the entire *m/z* range. This is a characteristic of the SCIEX equipment. Thus, even if diagnostic ions were observed with an intensity of 11 or below, MS-DIAL categorized them as "undetected." Furthermore, MS-DIAL defined the intensity of the lowest intensity ion observed in the product-ion spectrum as the "minimum peak intensity." If the intensity of a diagnostic ion was less than or equal to twice the minimum peak intensity (i.e., 2.0 x minimum peak intensity), it was classified as "undetected" despite the matched *m/z* value to that of diagnostic ions. This criterion was utilized to prevent the overestimation of lipid structure description. Based on these criteria, the ion of *sn1*-22:6 (*m/z* 705.509), illustrated in Fig. 2a, was classified as "undetected."

The structural diversity of the C = C positional isomers generated in MS-DIAL is inherently limited, and the configuration is seemingly optimized for mammalian cells. For monounsaturated fatty acids (MUFAs) with *O*-acyl and *N*-acyl chains, the potential C = C positions were derived from those listed in the LIPID MAPS Structure Drawing Tool for glycerophospholipid structures. Positions defined as multiples of three from the omega terminus, i.e., omega 3, 6, and 9, were included as potential sites. The C = C positions of sphingoid bases were obtained using the candidate list from the LIPID MAPS tool for sphingolipids, including delta 4, 6, 8, and 14, as the reference. Of these, the candidates of delta 6 were excluded because the C = C position was unusual in mammalian cells. Furthermore, only the delta 4 position is considered when the sphingoid base contains one double bond. For PUFAs not included in LIPID MAPS, candidate structures of omega 3, 6, and 9 fatty acids, including VLC-PUFA, were generated. The current version of MS-DIAL is not optimized for discovering new C = C positions; rather, it is designed to identify the most plausible candidate structures from a range of known double bonds and hydroxyl positions recognized in lipid biology. Naturally, the program can be customized for other species, such as plants and microorganisms, by modifying the range of double-bond positions defined in the XML format in the source code. Although it is recognized that the current capabilities of EAD-MS/MS techniques may not be sufficient for untargeted approaches, the development of structure annotation programs for interpreting EAD-MS/MS spectra remains crucial for advancing the standard-free structure elucidation of lipids. By integrating computational mass spectrometry techniques with targeted analyses, wherein lipid enrichment is followed by highly sensitive measurements, EAD-MS/MS can be leveraged to uncover new lipid structures.

## Elucidation of EAD-MS/MS spectra for glycerophospholipids in MS-DIAL

MS-DIAL provides an in-depth annotation pipeline for a wide range of glycerophospholipids. These include phosphatidylcholine (PC), phosphatidylethanolamine (PE), phosphatidylglycerol (PG), phosphatidylinositol (PI), phosphatidylserine (PS), bis(monoacylglycero)phosphate (BMP), lyso-type forms (LPC, LPE, LPG, LPI, and LPS), and plasmenyl/plasmanyl species (PC P-, PE P-, PC O-, and PE O-). While hemi-BMP (HBMP) and cardiolipin (CL) molecules were also examined, only molecular species-level annotations, such as CL 16:0_18:1_18:1_18:2, were feasible in EAD-MS/MS. This limitation was due to the poor sensitivity of diagnostic ions related to the C=C- and sn-positions, even with injections exceeding 1 pmol of the on-column volume on a conventional C18 analytical column. The annotation of the adduct forms $[M + H]^+$, $[M + NH_4]^+$, and $[M+Na]^+$ was supported by MS-DIAL, although we evaluated the annotation accuracy of the $[M + H]^+$ and $[M + NH_4]^+$ product ion spectra.

In the EAD-MS/MS spectra, the product ions of the polar head (PH; $X + H_2PO_4$), PH + $C_3H_4$, and PH + $C_2H_2O$, along with the neutral loss (NL) of the polar head (PH) group, are commonly observed in many glycerophospholipid subclasses. Here, "X" denotes the specific formula for each lipid subclass, for instance, $C_5H_{12}N^+$ for PC. Notably, the ion abundance of NL in the PC polar head group is typically lower than that in other phospholipids. Double-charged ions of the molecules were distinctly detected in PC and PE. The fragment ion from homolytic cleavage of the C1-C2 bond in the glycerol backbone, identified as "NL of sn1 + CH2," is detectable across most lipid subclasses. However, practical diagnosis using this ion is only viable for protonated PCs or sodium adduct phospholipids. The determination of sn-positional isomers is feasible for PC-O by confirming the fragment ion from the homolytic cleavage of the C1-C2 bond in the glycerol backbone. The distinction between PE-O and PE-P was as clear as in CID-MS/MS. Differentiation between PC-O and PC-P relies on scoring the C=C isomer candidates. For lysophospholipids (LPLs), sn1/sn2 isomer characterization is based on the neutral loss of $CH_2OH$, which is specific to sn2-LPL. Further details on the relationship between phospholipid structure and EAD-MS/MS spectra are available in Supplementary Fig. 2 and in the description of the lipidomics minimal reporting checklist[32] (Supplementary Note 1).

## Elucidation of EAD-MS/MS spectra for sphingolipids in MS-DIAL

This program provided an in-depth annotation pipeline for five sphingolipids: sphingomyelin (SM), ceramide (Cer), hexosylceramide (HexCer), dihexosylceramide (Hex2Cer), and sulfatide (SHexCer). Other sphingolipids, such as gangliosides and globosides have also been annotated at the molecular species level. EAD-MS/MS offers two distinct advantages over CID-MS/MS for the annotation of sphingolipids. First, the product ion of the sphingobase (SPB) and the neutral loss (NL) of the N-acyl chain was clearly observed, serving as crucial diagnostic markers to define lipids at the molecular species level, such as SM 18:1;O2/16:0. Second, the fragment ion resulting from the cleavage of each carbon-carbon bond containing a hydroxy moiety was distinctly detected, aiding in the annotation of OH positions in the sphingobase backbone. Consequently, EAD-MS/MS enhanced the OH-resolved sphingolipid profiling. Annotation of the C=C-position followed the same methodology as that used for glycerophospholipids. Further details on the relationship between sphingolipid structure and EAD-MS/MS spectra are provided in Supplementary Fig. 4 and the lipidomics minimal reporting checklist (Supplementary Note 1).

## Elucidation of EAD-MS/MS spectra for glycerolipids, sterols, and fatty acyls in MS-DIAL

The program offers the in-depth annotation pipeline for diacylglycerol (DG), triacylglycerol (TG), acylcarnitine (CAR), 1,2-diacylglyceryl-3-O-2′-(hydroxymethyl)-(N,N,N-trimethyl)-β-alanine (DGTA), 1,2-diacylglyceryl-3-O-4′-(N,N,N-trimethyl)-homoserine (DGTS), and their lyso forms (LDGTA and LDGTS). Monoacylglycerol (MG) was characterized at the molecular species level using EAD-MS/MS. The program also accommodates the detailed structural elucidation of fatty acyl esters of hydroxy fatty acids (FAHFA) and free fatty acids derivatized with 2-dimethylaminoethylamine (DMED), whose detailed methodology includes sample preparation and lipid enrichment will be reported elsewhere. For DG and TG, the sn1/sn2 positional isomers were determined from the EAD-MS/MS spectra of the sodium adducts. In contrast, the characterization of C=C positional isomers involves scoring candidates based on the reverse dot product similarity value between the experimental and in silico spectra related to acyl chains, as described above. Spectral information on DGTS and DGTA was obtained from cultivating two algal species: *Chlamydomonas reinhardtii*, which predominantly produces DGTS, and *Fistulifera solaris*, which generates DGTA. Cells were cultured according to a previously established protocol[33,34]. The distinction between DGTS and DGTA isomers, which was not feasible in CID, was enabled in EAD by comparing the ion abundance ratios of $m/z$ 204.123 ($C_9H_{18}NO_4$) and $m/z$ 236.149 ($C_{10}H_{22}NO_5$). Notably, the ion abundance at $m/z$ 236.149 surpassed that at $m/z$ 204.123 in DGTS, whereas this ratio was inverted in DGTA. Details of the EAD-MS/MS spectra of glycerolipids are available in Supplementary Fig. 5, as well as the description of the lipidomics minimal reporting checklist (Supplementary Note 1).

## Evaluation of calibration curve using DLPC and PAPC

The authentic standards 1,2-dilinoleoyl-sn-glycero-3-phosphocholine (DLPC) and 1-palmitoyl-2-arachidonoyl-sn-glycero-3-phosphocholine (PAPC) were purchased from Avanti Polar Lipids. Each compound was dissolved using 1:1 MeOH:CHCl₃ (v/v). A series of dilutions were prepared at concentrations of 10, 5, 2, 1, 0.5, 0.2, 0.1, 0.05, 0.02, and 0.01 μM for each compound. Given that 1 μL of each sample was injected, the on-column volume for the LC-MS method utilized in this study is estimated at 10,000 fmol, 5000, 2000, 1000, 500, 200, 100, 50, 20, and 10. The LC-MS conditions employed were mostly identical to those detailed in "*Investigating the kinetic energies to acquire information-rich EAD spectra of small molecules.*" Nine fragmentation conditions were explored during the targeted MS/MS scanning mode, including one CID setting (45 eV) and various EAD settings with different kinetic energies (8, 10, 12, 14, 16, 18, and 20 eV KE). A constant CE spread of 0, collision energy of 10 for EAD, and a time-of-flight (TOF) start mass of 170 were applied. For CID, a constant CE spread of 15 was applied. The ion source temperature was maintained at 250 °C, while other parameters were consistent with those described above. Each sample was analyzed three times (technical replicates = 3). The peak heights from the product ion chromatogram peak tops were used for quantification.

The peak heights of the diagnostic ions used to determine the sn-position and C=C-position of the lipids were investigated for DLPC and PAPC. For DLPC, the peak heights for product ions at $m/z$ 184.073 (mandatory to define the PC lipid subclass), $m/z$ 489.321 (NL of sn1 + CH₂; diagnostic ion to define the sn-position), $m/z$ 630.413 (H-loss at 18:2 C7 "C = C high" peak), $m/z$ 670.444 (H-loss at 18:2 C10 "C = C high" peak), $m/z$ 685.468 (radical at 18:2 C11 "C = C high" peak), and $m/z$ 725.499 (radical at 18:2 C14 "C = C high" peak) were investigated. For PAPC, the peak heights for product ions at $m/z$ 184.073 (mandatory to define the PC lipid subclass), $m/z$ 465.321 (NL of sn1-20:4 + CH₂; diagnostic ion to define the sn1 20:4), $m/z$ 513.321 (NL of sn1-16:0 + CH₂; diagnostic ion to define the sn1 16:0), $m/z$ 550.350 (H-loss at 20:4 C3 "C = C high" peak), $m/z$ 590.382 (H-loss at 20:4 C6 "C = C high" peak), $m/z$ 592.397 (H-gain at 20:4 C6 "C = C PUFA high peak"), $m/z$ 605.405 (radical at 20:4 C7 "C = C high" peak), $m/z$ 630.413 (H-loss at 20:4 C9 "C = C high" peak), $m/z$ 632.429 (H-gain at 20:4 C9 "C = C PUFA high peak"), $m/z$ 645.436 (radical at 20:4 C10 "C = C high" peak), $m/z$ 670.444 (H-loss at 20:4 C12 "C = C high" peak and H-gain at 16:0 C8), $m/z$ 685.468 (radical at 20:4 C13 "C = C high" peak), and $m/z$ 725.499

(radical at 20:4 C16 "C = C high" peak and radical at 16:0 C12), were examined. In this study, the product ion's peak was recognized as "not detected" if the peak height was zero in two of three samples.

## Evaluation of annotation results of co-eluted lipid molecules using a mixture of DLPC and PAPC

Two sets of the mixtures were prepared. The first set, termed "DLPC fixed," comprised seven mixtures containing DLPC and PAPC, where the DLPC concentration was consistently maintained at 1 μM, while PAPC concentrations varied at 0.1, 0.2, 0.5, 1.0, 2.0, 5.0, and 10 μM. The second set, termed "PAPC fixed," similarly consisted of seven mixtures. Here, the PAPC concentration was fixed at 1 μM, with DLPC concentrations adjusted to 0.1, 0.2, 0.5, 1.0, 2.0, 5.0, and 10. Each sample was analyzed three times (technical replicates = 3). The same mass spectrometer conditions described in the previous section were used. Flow injection, involving no column installation, was used to ensure the co-elution of the two metabolites. The other LC conditions were consistent with those previously described (see the section of "*Evaluation of calibration curve using DLPC and PAPC*"). The spectra of the co-eluted lipids were elucidated using MS-DIAL.

## Evaluation of MS-DIAL program by using LightSPLASH mixture

The LightSPLASH mixture (https://avantilipids.com/product/330732) containing 13 authentic lipid standards at 100 μg/mL each was purchased from Avanti Polar Lipids (Supplementary Data 3). This mixture was initially diluted fivefold with a 1:1 CHCl₃:MeOH (v/v) solvent. This diluted solution served as the starting point for a subsequent series of dilutions. The initial mixture was further diluted by factors of 2, 5, 10, 20, 50, 100, 200, 500, and 1000, using the same 1:1 CHCl₃:MeOH solvent. Each sample was analyzed thrice using LC-EAD (KE 14)-MS/MS. The lipid separation was carried out with the column of Unison UK-C18 MF (50 × 2.0 mm, 3 μm, Imtakt Corp., Kyoto, Japan) and the mobile phases of (A) acetonitrile (ACN):MeOH:H₂O (1:1:3, v/v/v) and (B) ACN:IPA (1:9, v/v). Both the solvents contained 10 nM ethylenediaminetetraacetic acid and 5 mM ammonium acetate. The injection volume, flow rate, sample rack temperature, and column oven temperature were set to 1 μL, 300 μL/min, 4 °C, and 45 °C, respectively. The gradient condition is as follows: 0.1% (B) (1 min), 0.1–40% (B) (4 min), 40–64% (B) (2.5 min), 64–71% (B) (4.5 min), 71–82.5% (B) (0.5 min), 82.5–85% (B) (6.5 min), 85–99.9% (B) (0.1 min), 99.9% (B) (1.4 min), 99.9–0.1% (B) (0.1 min), 0.1% (B) (4.4 min). A data-dependent MS/MS acquisition mode, called information-dependent acquisition in SCIEX, was used. The conditions for the EAD are as follows: MS1 scan range, *m/z* 70-1250; MS/MS scan range, *m/z* 150-1250; MS1 accumulation time, 200 ms; MS2 accumulation time, 100 ms; electron beam current, 7000 nA; ETC%, 100%; TOF start mass, 150; KE, 14 eV; CE, 10 V; CES, 0 V; ion source gas 1, 40; ion source gas 2, 80; curtain gas, 30; CAD gas, 7; temperature, 250; spray voltage, 5500; declustering potential, 80. Mass calibration was automatically performed using a SCIEX calibration delivery system. The mass spectra were analyzed using MS-DIAL version 5. The WIFF format files were directly imported into MS-DIAL. The following parameters were selected from the measurement setting page view: ionization mode, soft ionization, fragmentation method, EIEIO, target omics, lipidomics, MS1, centroid, MS2, and centroid. Details of the other parameter settings are listed in Supplementary Data 4. In this evaluation, the representative annotation was determined as follows. If the same lipid name was annotated in at least two of the three replicates, that name was used as the representative annotation. If the annotation results differed across all three replicates, the lipid with the highest score was adopted as representative.

## Evaluation of MS-DIAL program by using a mixture of UltimateSPLASH and in-house lipid standards

A standard mixture was prepared to evaluate the performance of the MS-DIAL algorithm for the EAD spectral annotation (Supplementary

Data 5). The Ultimate SPLASH, containing 69 lipid molecules, was purchased from Avanti Polar Lipids. The concentrations of the compounds in the original solutions varied from 26.87 μM 192.5 μM. This solution was diluted by factors of 2, 5, 10, 20, 50, 100, and 200 in 1:1 MeOH:CHCl₃ (v/v). Furthermore, an in-house mixture containing 41 lipid standards, previously employed in a different study[35], was also utilized, where each lipid was adjusted to a concentration of 50 μM. The in-house mixture contained the same amount of EquiSPLASH (64 times diluted from the original material) as the internal standards, although the lipids were not evaluated because the deuterium isotope was in the acyl chains. This solution was subjected to the same dilution process as that used for UltimateSPLASH. Nineteen lipids were excluded from the evaluation, two of which (SM 18:1;O/18:1-d9 and CE-d7 18:1) with the same *m/z* value as the standards in EquiSPLASH. The other lipids, including FA 20:4, SPB 18:1;O2, PE N-FA 20:4 18:1/18:1, PA 18:1/18:1, LPA 18:1, HBMP 18:1/18:1/18:1, cholesterol, CL 16:0_18:1/16:0_18:1, SM 18:1;O2/24:1(FA 16:0), and gangliosides, including GD1b, GD3, NGcGM3, GT1b, GM1, GQ1b, and GD1a, were excluded because the detection of these metabolites is hard in positive ion mode and beyond the scope of this study.

A leaf lipid extract from uniformly ¹³C-labeled (>97 atom % ¹³C) *Nicotiana tabacum* was used as the background matrix. The plant materials were purchased from IsoLife (Wageningen, Netherlands). The lipid extraction protocol was performed according to a previous study[30]. Briefly, the plant material in a 2.0 mL microcentrifuge tube was milled by shaking at 900 rpm for 3 min on a Shake Master Neo (BMS, Tokyo, Japan) using zirconia beads. From the frozen powdered plant material, 5 mg was measured and transferred into a new 2.0 mL tube. To the tube, 1 mL of a solvent mixture consisting of 5:2:2 MeOH:H₂O:CHCl₃ (v/v/v) was added. After stirring on a vortex mixer vigorously, the homogenate was incubated for 30 min at 1200 rpm at 25 °C, followed by the addition of 400 μL of H₂O for liquid-liquid separation. Twenty microliters of the bottom solvent layer were transferred into a new 2.0 mL tube, where a total of eight tubes were prepared. Fifty microliters of each dilution ratio from the dilution series of Ultimate SPLASH and in-house standard mixture solutions were added to each tube. A total of 100 μL of the solvent used to create the dilution series, namely a 1:1 MeOH:CHCl₃ (v/v) solvent, was added to the remaining tube. The samples were dried with a vacuum dryer and resuspended in 50 μL of MeOH, including 1 μL of EquiSPLASH mixture. The LC-MS/MS settings, MS-DIAL settings, and evaluation methods were the same as those described in the previous section.

## Evaluation of annotation results of co-eluted lipid molecules using a mixture of PC 16:0/18:1(9), PC 16:0/18:1(11), and PC 18:1(9)/16:0, mouse brain, and NIST SRM 1950 plasma

Avanti IsoPure standards PC 16:0/18:1(9Z), PC 16:0/18:1(11Z), and PC 18:1(9Z)/16:0 were purchased. The purity of these IsoPure standards was confirmed by C13 NMR with less than 5% acyl migrated.

The retention times of these lipids were confirmed using the same LC condition used for LightSPLASH and UltimateSPLASH mixtures analyses. Because the retention times of these lipids were mostly equivalent and not differentiated (Supplementary Fig. 7a), a LC condition where the injection-to-injection time was set to approximately 9 minutes was used to increase the throughput of the following experiments described in this section. The conditions for chromatography were based on our previous study[36]. Briefly, the same Imtakt UK-C18 MF column described above was used. The column was maintained at 65 °C and a flow rate of 0.6 mL/min. The same mobile phases, A and B, were used and a sample volume of 1 μL was used for the injection. The separation was performed using the following gradient: 0 min 0.1% (B), 0.1 min 0.1% (B), 0.2 min 15% (B), 1.1 min 30% (B), 1.4 min 48% (B), 5.6 min 82% (B), 6.9 min 99% (B), 7.1 min 99% (B), 7.2 min 0.1% (B), and 8.6 min 0.1% (B). The other LC conditions were the same as described in the above section. The targeted MS/MS scanning

mode for the precursor *m/z* of the protonated form of PC 16:0_18:1 was used. The TOF mass range was set from a starting mass of 150 to an end mass equal to the precursor *m/z*. Three fragmentation conditions were explored: CID 40 V with a CE spread of 15 V, EAD KE 10 eV, and EAD KE 14 eV. The other parameters were the same as described previously. For the OAD analysis, a quadrupole/time-of-flight mass spectrometer system LCMS-9030 equipped with OAD RADICAL SOURCE I (Shimadzu, Kyoto, Japan) was employed following the previously described protocol[7]. The targeted MS/MS scanning mode was used for the precursor *m/z* of the protonated form of PC 16:0_18:1 in positive ion mode. The parameters were MS1 accumulation time, 200 ms; OAD-MS/MS accumulation time, 100 ms; collision energy, 10 eV; nebulizer gas flow, 3 L/min; interface temperature, 300 °C, and interface voltage, +4.0.

Calibration curves were constructed for each of the authentic standards to assess the linear range of MS1 and product ion peaks (Supplementary Fig. 7c). The concentration of each molecule was adjusted to 10 nM, 20 nM, 50 nM, 100 nM, 200 nM, 500 nM, 1000 nM, 2000 nM, and 5000 nM with 1% of EquiSPLASH. A procedure blank containing the same amount of EquiSPLASH internal standards was also prepared. Three technical replicates were prepared. The injection volume was set to 1 µL; the same injection volume was used for subsequent experiments. Standard mixtures of varying concentrations were prepared for the pairs of PC 16:0/18:1(9Z) and PC 16:0/18:1(11Z), as well as PC 16:0/18:1(9Z) and PC 18:1(9Z)/16:0. The final concentrations of the pair of (A) PC 16:0/18:1(9Z) and (B) PC 16:0/18:1(11Z) were adjusted to A:B = 1000 nM:100 nM, A:B = 750 nM:250 nM, A:B = 500 nM:500 nM, A:B = 250 nM:750 nM, and A:B = 100 nM:1000 nM. Three technical replicates were prepared. Lipid extraction from NIST SRM 1950 plasma was performed following a previously described protocol[10]. Briefly, 10 µL of plasma sample was mixed with 1,000 µL of an ice-cold MeOH/CHCl$_3$/H$_2$O (10:4:4, v/v/v) solvent. Lipids were extracted using a vortex mixer for 1 min and subjected to ultrasonication for 5 min. The solution was centrifuged at 16,000 × *g* for 5 min at 4 °C, and 700 µL of the supernatant was transferred into a clean tube. The supernatant was mixed with 235 µL of CHCl$_3$ and 155 µL of H$_2$O using a vortex mixer for 1 min. After centrifugation at 16,000 × *g* for 5 min at 4 °C, 44.6 µL of the organic (bottom) layer was collected. Finally, lipid extracts were dried using a centrifugal evaporator.

For brain samples, the animal experiments were performed in accordance with the ethical protocol approved by the Tokyo University of Agriculture and Technology (R5-50). Nine-week-old C57BL/6 J male mice were purchased from SLC (Shizuoka, Japan). The mice were housed in 12 h light-dark cycle under conventional conditions, and controlled atmospheric conditions (temperature; 20–22 °C, humidity; 40–60%) with free access to food and water. The mice were fed the chow of CE-2 (CLEA Japan, Tokyo, Japan) for 2 weeks. The eye and brain organs were harvested and immediately frozen after dissection and stored at −80 °C until lipid extraction. The lipid extraction protocol of the eye organ was described in the next section. Samples were lyophilized prior to lipid extraction. The samples were homogenized using a mixer mill (MM 301; Retsch, Germany) at 20 Hz for 2 min. The mixer mill rack was pre-chilled with liquid nitrogen prior to homogenization. A 4 mg sample of the mouse brain was subjected to the Bligh and Dyer method. Samples were mixed with 1000 µL of an ice-cold MeOH/CHCl$_3$/H$_2$O (10:4:4, v/v/v) solvent. Lipids were extracted using a vortex mixer for 1 min and subjected to ultrasonication for 5 min. The solution was centrifuged at 16,000 × *g* for 5 min at 4 °C, and 700 µL of the supernatant was transferred into a clean tube. The supernatant was mixed with 235 µL of CHCl$_3$ and 155 µL of H$_2$O using a vortex mixer for 1 min. After centrifugation at 16,000 × *g* for 5 min at 4 °C, 10 µL of the organic (bottom) layer was collected. Finally, lipid extracts were dried using a centrifugal evaporator. Based on the dilution series data of mouse brain lipid extract and the calibration curves

of PC isomers, an extract with PC 16:0_18:1 concentration at 1000 nM was prepared. Subsequently, PC 16:0/18:1(8) purchased from Avanti Polar Lipids was spiked at 100 nM, 500 nM, and 1000 nM. The same LC-MS/MS condition described in this section was used for the biological samples.

## Characterization of very long chain PUFA (VLC-PUFA) containing PC in the eye tissue of mice

For the lipid extraction, an incision was made in the mouse eye with scissors, and a single 5-mm diameter zirconia bead was inserted. All procedures were performed on ice. The samples were homogenized using a mixer mill (MM 301; Retsch, Germany) at 20 Hz for 2 min. The mixer mill rack was pre-chilled with liquid nitrogen prior to homogenization. Methanol was added to the sample tube. After the mixture was vortexed for 10 s, an appropriate amount of MeOH (<100 µL) solving 0.5 mg of eye tissue weight was transferred to a 2 ml tube. Methanol (MeOH), chloroform (CHCl$_3$), and H$_2$O were added to the tube to achieve a final solvent mixture of 1,000 µL MeOH/CHCl$_3$/H$_2$O (10:4:4, v/v/v). Lipids were extracted using a vortex mixer for 1 min and then ultrasonicated for 5 min. The solution was centrifuged at 16,000 × *g* for 5 min at 4 °C, and 700 µL of the supernatant was transferred to a clean tube. The supernatant was mixed with 235 µL of CHCl$_3$ and 155 µL of H$_2$O using a vortex mixer for 1 min. After a subsequent centrifugation at 16,000 × *g* for 5 min at 4 °C, 330 µL of the organic (bottom) layer was collected. Finally, lipid extracts were dried using a centrifugal evaporator.

The eye samples were analyzed with and without enrichment for VLC-PUFA PC molecules. The enrichment process was performed as follows: A dried lipid extract was dissolved by applying 60 µL of MeOH with 1% formic acid. A MonoSpin Phospholipid (GL Sciences Inc., Tokyo, Japan) solid-phase extraction (SPE) column was activated with 200 µL of MeOH with 1% formic acid, followed by the application of 50 µL of the sample. Centrifugation during SPE was performed at 3000 × *g* for 1 min. After washing the SPE column with 200 µL of 100% MeOH, the phospholipids were fractionated twice using 200 µL of 9:10:1 IPA:H$_2$O:NH$_3$ (v/v/v) solvent[37]. After solvent evaporation using a vacuum dryer, the residue was resuspended in 50 µL of 95% MeOH and further fractionated using a MonoSpin C18 SPE column (GL Sciences Inc., Tokyo, Japan). The column was conditioned with 200 µL of 100% MeOH followed by 200 µL of H$_2$O. Subsequently, 50 µL of the sample was applied. The column was then washed with 200 µL of H$_2$O, 200 µL of 95% MeOH, and twice with 200 µL of 96.5% MeOH. Finally, the VLC-PUFA PC fraction was eluted using 200 µL of 100% MeOH and 200 µL of 1:1 MeOH:CHCl$_3$, with the solvent subsequently evaporated using a vacuum dryer. These processes were omitted from analyses without lipid enrichment. After the sample was dissolved in 50 µL of 100% MeOH containing 1 µL of EquiSPLASH and 1 µM FA 16:0-d3 and FA 18:0-d3, it was transferred to an LC-MS vial. Four biological replicates were analyzed. For the lipidome profiling, the same LC-MS/MS conditions used for the evaluation of the MS-DIAL program were used, and 1 µL from each vial was injected. Samples without lipid enrichment were analyzed using the DDA method of EAD 14 eV KE. The same MS settings as those described for the LightSPLASH and UltimateSPLASH analyses were used for the EAD, while the LC gradient condition was slightly different. The gradient condition is as follows: 0.5% (B) (1 min), 0.5–40% (B) (4 min), 40–64% (B) (2.5 min), 64–71% (B) (4.5 min), 71–82.5% (B) (0.5 min), 82.5–85% (B) (6.5 min), 85–99% (B) (1.0 min), 99.9% (B) (2.0 min), 99.9–0.1% (B) (0.1 min), 0.1% (B) (4.9 min). This experiment is also described in the lipidomics minimal reporting checklist (Supplementary Note 2).

## Database creation of species/tissue-specific m/z and collision-cross section values of lipids

The database for the lipid annotation of matrix-assisted laser desorption/ionization (MALDI) coupled with tapped ion mobility mass

spectrometry (TIMS) data was prepared as follows: Conventional reverse-phase LC-MS/MS-based lipidomics data from MetaboBank ID MTBKS215, MTBKS216, and MTBKS217 were downloaded from the website (https://www.ddbj.nig.ac.jp/metabobank/index.html). The dataset contained 136 unique biological origins, including 27 unique tissues or cell types from mice (C57B6J or C57B6N), two cell types and human plasma, 99 unique pairs of plant species/tissues, and 7 algae species. The *m/z* and collision cross-section (CCS) database of lipids that were characterized in a specific biospecimen was created for each of the 136 unique biological origins. The full list of the characterized lipid molecules in each biological study is available in Supplementary Data 9, where an average of 236 lipid molecules per sample were recorded. The CCS values of $[M + H]^+$, $[M + NH_4]^+$, $[M+Na]^+$, $[M-H]^-$, $[M + HCOO]^-$, $[M + CH_3COO]^-$, $[M + H-H_2O]^+$, $[M + K]^+$, and $[M+Li]^+$ were predicted by the machine learning model created in the previous study with a small modification. Briefly, the experimental CCS values of 3601 ion forms of 2799 molecules from 95 lipid subclasses were used for the model development, where the training data set is available in the supplementary data of a previous report[10]. The descriptors and fingerprints of the molecular structure were calculated by NCDK v1.5.6 (https://kazuyaujihara.github.io/NCDK/html/e2ff06cc-99b7-4f8b-95c5-53965548639f.htm). With these variables, the XGBoost function optimized using the parameter tuning method of PicNet.XGBoost (v0.2.1; https://www.nuget.org/packages/PicNet.XGBoost/), was used to create the CCS prediction model.

### Data processing of mass spectrometry imaging data

Mass spectrometry imaging data of the eye tissues from C57B6/J and acyl-coenzyme A (CoA) synthetase (ACSL) 6 knockout mice were downloaded from the RIKEN DROPMet website (http://prime.psc.riken.jp/menta.cgi/prime/drop_index), identified under index number DM0048. This dataset was obtained using the "TIMS-ON" mode, indicating that ion mobility separation was executed. In this study, only positive ion mode data generated using a matrix of DHB (2,5-dihydroxybenzoic acid) were analyzed. The detailed methodologies are available in a previous paper[22]. Timsdata.dll is necessary for reading the Bruker raw data files were downloaded from the Bruker SDK website (https://www.bruker.com/en/services/software-downloads.html). The data structure encompasses approximately 400 spectra for each MALDI spot, expandable along the drift time and *m/z* axis. The initial step in the algorithm involves accumulating all spectra from all MALDI spots, applying binning values of *m/z* 0.005 and drift time 0.01 ms, adjustable by users. The accumulated spectral data were stored in an intermediate file. The peak picking algorithm of MS-DIAL was performed for the accumulated spectra, resulting in the generation of peak features defined by *m/z*, drift time, collision cross section (CCS), and peak height. Lipid annotation was performed using the above database, which contained the *m/z* and CCS reference values of lipids detected in the eye tissues of mice. The tolerances for *m/z* and CCS were set to 0.01 Da and 20 Å², respectively. Peak features were utilized to map the ion distributions in the spatial images. For the ion abundance mapping into each of the MALDI spot pixels, the ions with tolerances of 0.01 Da and 5 Å² from the experimental values of peak features were accumulated, and the integrated data is stored as an intermediate file in the MS-DIAL application.

### Re-analysis of publicly available untargeted lipidomics data analyzing the eye tissue of Acsl6 KO mouse

The untargeted lipidomics data using a reverse-phased chromatography method was downloaded from the RIKEN DROP Met website (http://prime.psc.riken.jp/menta.cgi/prime/drop_index), identified under index number DM0048, which is the same as above. The negative-ion mode data were analyzed using MS-DIAL 5, and the parameters used are listed in Supplementary Data 10.

### HeLa cells experiment with the addition of VLC-PUFA

The omega-3 VLC-PUFA compound 14*Z*,17*Z*,20*Z*,23*Z*,26*Z*,29*Z*-dotriacontahexaenoic acid (FA 32:6; ID: CAY10632) was purchased from Cayman Chemicals. HeLa cells (RIKEN BRC: Cell No. RCB3680) were maintained at 37 °C in DMEM (Dulbecco's modified Eagle medium) supplemented with 10% EquaFETAL (Atlas Biologicals, Inc.) and 1% penicillin-streptomycin solution (Fujifilm, Wako, Japan) with 5% CO₂. For sample preparation for glycerolipid and glycerophospholipid profiling, cells ($3 \times 10^5$ cells per well) were incubated in 6-well plates (Thermo Scientific, Nunc, Denmark) for 3 h. For LPA analysis, cells ($5 \times 10^5$ cells/well) were incubated in 10 cm dishes (TPP, Switzerland). VLC-PUFA (FA 32:6) dissolved in 0.4% ethanol was added to the well plate at final concentrations of 1, 10, and 40 μM, with four biological replicates per condition. A solution of 0.4% ethanol was used as the vehicle control. After incubation for 24 h, the medium was removed, and the cells were washed twice with ice-cold phosphate-buffered saline (PBS) without calcium and magnesium. To profile the glycerolipids and glycerophospholipids, the cells were detached using a cell scraper with 1000 μL of ice-cold MeOH. The solvent (700 μL) was then transferred to a tube. In addition, 400 μL of ice-cold MeOH was added to each well plate, and 300 μL of the solvent was transferred to the same tube. To profile LPA, ice-cold PBS was used instead of ice-cold MeOH. The PBS solution was discarded after centrifugation at $16,000 \times g$ for 5 min at 4 °C, and the cell pellet was stored at −80 °C until LPA analysis (see next section).

The solvent was sonicated in BIORUPTOR II (CosmoBio, Tokyo, Japan) for 10 cycles, each taking 0.5 min for sonication and 0.5 min to maintain the water temperature at 4 °C. After adding 400 μL of CHCl₃, lipids were extracted using a vortex mixer for 1 min and ultrasonication for 5 min. The solution was centrifuged at $16,000 \times g$ for 5 min at 4 °C, and 700 μL of the supernatant was transferred to a clean tube. After adding 300 μL of CHCl₃ and 400 μL of H₂O to the tube, the solution was vortexed for 1 min and ultrasonicated for 5 min. After a subsequent centrifugation at $16,000 \times g$ for 5 min at 4 °C, 400 μL of the organic (bottom) layer was collected. Lipid extracts were dried using a centrifuge evaporator. After the sample was dissolved in 60 μL of 100% MeOH containing 1 μL of EquiSPLASH and 1 μM FA 16:0-d3 and FA 18:0-d3, it was transferred to an LC-MS vial. The lipids were analyzed using ESI(+)- and ESI(-)-CID DDA modes. The mass spectrometer settings for the CID mode were as follows: MS1 and MS2 mass ranges, *m/z* 70–1250; MS1 accumulation time, 200 ms; Q1 resolution, units; MS2 accumulation time, 50 ms; maximum candidate ions, 10; CAD gas, 7; intensity threshold for DDA, 10 cps; dynamic background subtraction, ticked; and no inclusion or exclusion lists were used. The following settings were used for positive/negative ion mode, independently: ion source gas 1, 40/50 psi; ion source gas 2, 80/50 psi; curtain gas, 30/35 psi; source temperature, 250/300 °C; spray voltage, 5500/-4500 V; declustering potential, 80/−80 V; and collision energy, 40/-42 ± 15 eV. The LC condition is the same as used for eye-lipidome analysis. This experiment is also described in the lipidomics minimal reporting checklist (Supplementary Note 3).

### LC-MS/MS analysis for lysophosphatidic acid (LPA) profiling for HeLa cells

Methyl tert-butyl ether (MTBE) and trimethylsilyl (TMS)-diazomethane were purchased from Sigma-Aldrich (Tokyo, Japan) and Tokyo Chemical Industry (Tokyo, Japan), respectively. LPA analysis was performed using a modified protocol from a previous study[38]. Next, 200 μL ice-cold MeOH with 0.1% formic acid was added to the HeLa cell pellet. The solvent was sonicated in BIORUPTOR II (CosmoBio, Tokyo, Japan) for 10 cycles, each taking 0.5 min for sonication and 0.5 min to maintain the water temperature at 4 °C. After centrifugation at $16,000 \times g$ for 5 min at 4 °C, 190 μL of the supernatant was transferred into a new tube and dried by a centrifugal evaporator. The sample was dissolved with 120 μL of MeOH containing 0.5 μM of LPA 17:1 as the

internal standard. For the derivatization, 50 μL of 2 M TMS-diazomethane was added and incubated for 20 min at 25 °C by 800 rpm in Ballerina NSD-12J (Tokyo Garasu Kikai Co., Ltd., Japan). After adding 3 μL of acetic acid, 400 μL of MTBE and 100 μL of H$_2$O were added. After vortex mixing at the maximum speed for 5 min at 25 °C in Ballerina NSD-12J, 360 μL of the supernatant was collected and dried up by centrifugal evaporator. The sample was resuspended in 30 μL MeOH containing 1 μL EquiSPLASH and 1 μM FA 16:0-d3 and FA 18:0-d3. LPA analysis was performed using the MRMHR mode, targeting the protonated form of bismethyl LPA (BisMeLPA) 32:6 under the mostly same LC-MS conditions described in the previous section. The mass spectrometer settings were as follows: MS1 mass range, *m/z* 100–1000; MS1 accumulation time, 250 ms; Q1 resolution, units; MS2 accumulation time, 100 ms. The other settings are the same as described above. This experiment is also described in the lipidomics minimal reporting checklist (Supplementary Note 4).

### GPAT1 protein reconstitution using a wheat germ cell-free synthesis system

GPAT recombinant proteins were prepared using a cell-free system according to a previously reported protocol[24]. Complementary deoxyribonucleic acid (cDNA) encoding human GPAT1, also known as GPAM (glycerol-3-phosphate acyltransferase, mitochondrial), was cloned into the pEU vector (CellFree Science, Japan). GPAT1 was synthesized by Integrated DNA Technologies (Coralville, IA, US). Hereafter, the native gene is shown as GPAT1$^{WT}$. The gene arrays and sequence details are listed in Supplementary Data 11. The cDNA encoding the mutant GPAM$^{H230A}$ was generated by site-directed mutagenesis PCR, according to the manufacturer's protocol (TaKaRa, PrimeSTAR Mutagenesis Basal Kit, Japan). The primers used for cloning in this study are listed in Supplemental Data 11. The native pEU vector was used as vector control, termed "Mock." Protein reconstitution with liposomes was performed using the WEPRO7240 Expression Kit and Asolectin Liposomes (Cell Free Sciences, Japan) according to a previous study[24]. The presence of the synthesized proteoliposomes was verified by sodium dodecyl sulfate-polyacrylamide gel electrophoresis (SDS-PAGE) and Coomassie brilliant blue (CBB) staining (Supplementary Fig. 12). The amount of the expressed proteins was determined using a bovine serum albumin (BSA) standard, and 1 μg of expressed protein-containing liposomes was used in the enzymatic assays. The average liquid volume from enzymatic assays involving GPAT1$^{WT}$ and GPAT1$^{H230A}$ was utilized for the mock sample's enzymatic analysis.

### Evaluation of GPAT1 enzyme activity

GPAT1 enzymatic activity was determined as in the previous study[39]. The assay was performed for 1 h at 37 °C in 100 μL solution containing 1 μg of protein-containing proteoliposomes, 75 mM Tris-HCl (pH 7.5), 4 mM MgCl$_2$, 1 mg/mL BSA (essentially fatty acid-free), 500 μM CoA, 2.5 mM ATP, 8 mM NaF, 800 μM glycerol 3-phosphate, and 10 μM of a fatty acid. In this study, three fatty acids, palmitic acid (uniformly $^{13}$C-labeled, U-$^{13}$C), DHA, and FA n-3-32:6, were examined. A solution of 1% ethanol was used as the vehicle control. The reaction mixture was incubated for 10 min at 37 °C. The SPE method using MonoSpin C18 SPE column (GL Sciences Inc., Tokyo, Japan) was used for lipid extraction. First, 100 μL of 1 M ammonium acetate and 200 μL of MeOH containing 17:1 CoA (0.5 μM) and LPA 17:1 (0.5 mM) as internal standards were added to the reaction mixture. After the SPE column was activated with 200 μL MeOH, 200 μL H$_2$O, and 200 μL of 1 M ammonium acetate, 360 μL of the sample was applied. After the column was washed with 200 μL H$_2$O and 200 μL hexane, the targeted lipid fractions containing LPAs and acyl CoAs were retrieved using 200 μL MeOH. The solvent was dried and used for the LPA and acyl-CoA analyses.

### LC-MS/MS analysis for LPA and acyl CoA profiling for the extract from GPAM assay

The dried sample was dissolved in 90 μL MeOH, 30 μL of which was transferred to an LC-MS vial for acyl-CoA profiling. The derivatization and LC-MS/MS protocol described above for LPA analysis were performed using the remaining solvent. Details of LPA analysis are described in the lipidomics minimal reporting checklist (Supplementary Note 5). The acyl CoA separation was carried out with the column of L-column3 C8 (3 μm, 2.0 × 100 mm metal free, CERI, Japan), and the mobile phases of (A) MeOH:H$_2$O (1:4, v/v) with 0.05% NH$_3$ and (B) MeOH:ACN (1:4, v/v) with 0.05% NH$_3$. The injection volume, flow rate, sample rack temperature, and column oven temperature were set to 5 μL, 250 μL/min, 4 °C, and 40 °C, respectively. The gradient conditions were 0.1% (B) (1.2 min), 0.1–100% (B) (4.8 min), 100% (B) (4 min), 100–0.1% (B) (0.1 min), and 0.1% (B) (4.9 min). Agilent 1290 Bio UHPLC coupled with 6546 QTOF system was used for the LC-MS analysis. The MS settings were as follows: gas temperature, 325 °C; gas flow, 12 L/min; nebulizer (psig), 55; sheath gas temperature, 300 °C; sheath gas flow, 11 L/min; vcap, 3500 V; nozzle voltage, 1000 V; fragmentor, 175 V; skimmer 65 V; octupole RF Vpp, 750 V; MS1 and MS2 ranges, *m/z* 90-1250; isolation width, narrow (-1.3 *m/z*); and collision energy, 20 eV. The MS/MS spectra were acquired by targeted MS/MS scanning mode. The detail of acyl-CoA analysis is also described in the lipidomics minimal reporting checklist (Supplementary Note 6).

### Reporting summary

Further information on research design is available in the Nature Portfolio Reporting Summary linked to this article.

## Data availability

The spectral data of the authentic standards are available in MSP format files on the RIKEN DROPMet website (http://prime.psc.riken.jp/menta.cgi/prime/drop_index) under the index number DM0054, and the details are described in the readme file. All raw LC-MS data were available with the same index number as the RIKEN DROPMet. Spatial- and untargeted lipidomics data for eye tissues in mice are available under the index number DM0048. Source data are provided with this paper.

## Code availability

The MS-DIAL source code is available at https://github.com/systemsomicslab/MsdialWorkbench. The MS-DIAL tutorial and the demonstration data are available at https://systemsomicslab.github.io/msdial5tutorial/ and https://zenodo.org/communities/msdial, respectively.

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

## Acknowledgements

This study represents a portion of the dissertation submitted by Yuki Matsuzawa to the Tokyo University of Agriculture and Technology in partial fulfillment of the requirement for his Ph.D. This study was supported by the JSPS KAKENHI (21K18216, 24K02011, 24H00043, 24H00392, 24K21269 to Hiroshi T.), National Cancer Center Research and Development Fund (2023-A-08, Hiroshi T.), AMED Japan Program for Infectious Diseases Research and Infrastructure (21wm0325036h0001, Hiroshi T.), AMED Brain/MINDS (JP15dm0207001, Hiroshi T.), JST National Bioscience Database Center (JPMJND2305, Hiroshi T.), JST ERATO "Arita Lipidome Atlas Project" (JPMJER2101, M.A. and Hiroshi T.) and Technologically Advanced research through Marriage of Agriculture and engineering as Groundbreaking Organization (TAMAGO to J.M. and Hiroshi T.). T.H. was supported by the ATIP-Avenir program (CNRS/Inserm) and the Global Innovation Research funds of the Tokyo University of Agriculture and Technology.

## Author contributions

T.H., M.A. and Hiroshi T. designed this study. Hiroaki T., Manami T., Y. T., and Hiroshi T. performed the LC-MS/MS analyses. Y.M., Mikiko T., K.N., and Hiroshi T. developed MS-DIAL, and Manami T., K.N., Y.T., T.O., Y.K., S.K., Kanako T., B.B., M.K., and A.K. created the tutorial and contributed to MS-DIAL 5 improvements. U.T. provided technical support for the EAD-MS/MS experiments. M.H. and J.M. performed the mouse

experiments. N.S. performed the imaging MS data analysis. Manami T., T.H., K.S., M.M., and M.H.C. performed cloning and enzyme assays of GPAT proteins. H.U., Kana T., S.T., T.M., R.K., T.T., and T.Y. prepared authentic standards and biologically created definable lipid molecules. M.O. and Hidenori T. assisted in OAD-MS/MS experiments. Hiroshi T. wrote the manuscript, and T.H. improved the draft. All authors have thoroughly discussed this project and helped improve the manuscript.

## Competing interests

U.T. is an application specialist in ABSciex, Japan. K.T. and S.T. are the research scientists in AGC Inc., Japan. M.O. and Hidenori T. are research scientists at SHIMADZU CORPORATION, Japan. All the other authors declare no competing interests.

## Additional information

**Hiroaki Takeda** [1,2,16], **Yuki Matsuzawa**[1,16], **Manami Takeuchi**[1,16], **Mikiko Takahashi** [3], **Kozo Nishida**[1], **Takeshi Harayama** [4,5] ✉, **Yoshimasa Todoroki**[1], **Kuniyoshi Shimizu** [1], **Nami Sakamoto** [1], **Takaki Oka**[1], **Masashi Maekawa** [6], **Mi Hwa Chung**[1], **Yuto Kurizaki**[1], **Saki Kiuchi**[1], **Kanako Tokiyoshi** [1], **Bujinlkham Buyantogtokh**[1], **Misaki Kurata**[1], **Aleš Kvasnička** [7,8,9], **Ushio Takeda**[10], **Haruki Uchino** [6,11], **Mayu Hasegawa**[12], **Junki Miyamoto** [12], **Kana Tanabe**[13], **Shigenori Takeda** [13], **Tetsuya Mori**[3], **Ryota Kumakubo**[1], **Tsuyoshi Tanaka** [1], **Tomoko Yoshino** [1], **Mami Okamoto**[14], **Hidenori Takahashi** [14], **Makoto Arita** [6,11,15] ✉ & **Hiroshi Tsugawa** [1,3,5,11,15] ✉

[1]Department of Biotechnology and Life Science, Tokyo University of Agriculture and Technology, 2-24-16 Naka-cho, Koganei-shi, Tokyo 184-8588, Japan. [2]RIKEN Center for Brain Science, 2-1 Hirosawa, Wako, Saitama 351-0106, Japan. [3]RIKEN Center for Sustainable Resource Science, 1-7-22 Suehiro-cho, Tsurumi-ku, Yokohama, Kanagawa 230-0045, Japan. [4]Institut de Pharmacologie Moléculaire et Cellulaire, Université Côte d'Azur - CNRS UMR7275 - Inserm U1323, 660 Route des Lucioles, 06560 Valbonne, France. [5]Institute of Global Innovation Research, Tokyo University of Agriculture and Technology, 2-24-16 Naka-cho, Koganei-shi, Tokyo 184-8588, Japan. [6]Graduate School of Pharmaceutical Sciences, Keio University, Minato-ku, Tokyo 105-8512, Japan. [7]Faculty of Medicine and Dentistry, Palacký University Olomouc, Hněvotínská 3, 779 00 Olomouc, Czech Republic. [8]Laboratory for Inherited Metabolic Disorders, Department of Clinical Biochemistry, University Hospital Olomouc, Zdravotníků 248/7, 779 00 Olomouc, Czech Republic. [9]Department of Medical Biochemistry, Oslo University Hospital, Sognsvannsveien 20, 0372 Oslo, Norway. [10]K.K. ABSciex Japan, Shinagawa, Tokyo 140-0001, Japan. [11]RIKEN Center for Integrative Medical Sciences, 1-7-22 Suehiro-cho, Tsurumi-ku, Yokohama, Kanagawa 230-0045, Japan. [12]Department of Applied Biological Science, Tokyo University of Agriculture and Technology, 3-5-8 Saiwai-cho, Fuchu, Tokyo 183-8509, Japan. [13]Innovative Technology Laboratories, AGC Inc., 1-1 Suehiro-cho, Tsurumi-ku, Yokohama 230-0045, Japan. [14]Shimadzu Corporation, 1 Nishinokyo Kuwabara-cho, Nakagyo-ku, Kyoto 604-8511, Japan. [15]Graduate School of Medical Life Science, Yokohama City University, Yokohama, Japan. [16]These authors contributed equally: Hiroaki Takeda, Yuki Matsuzawa, Manami Takeuchi. ✉e-mail: harayama@ipmc.cnrs.fr; marita@keio.jp; htsugawa@go.tuat.ac.jp

