## [Transparent Peer Review file · Nature Communications]

MS-DIAL 5 multimodal mass spectrometry data mining unveils lipidome complexities

Corresponding Author: Dr Hiroshi Tsugawa

Version 0:

Reviewer comments:

Reviewer #1

(Remarks to the Author)

In this manuscript the authors present their work on the software MSDIAL5. MSDIAL has been an important software for the analysis of lipid mass spectra from various experimental setups including different mass spectrometers and recent technology such as TIMS-TOF. In this new iteration number 5 the authors focus on analyzing spectra from electron-activated dissociation (EAD) as provided by the ZenoTOF 7600 mass spectrometer. The authors mainly describe their newly-developed workflow for identifying molecular lipid species with defined carbon-carbon double bond (C=C) positions which is the primary novelty and advantage of EAD fragmentation. This is an important new feature added to MSDIAL 5 to increase the structural resolution in MS-based lipid identification. However, there are a few things that attention.

Major issues:

1. In the process of identifying the C=C the authors use a pattern +2 Da around a C=C position they call "V-shape". However, this definition is not exhaustive. The sheer existence of a V-shape around a particular position in the spectrum is not indicative of the existence of a C=C at that position. As an example, in Fig. 1b V-Shape the spectrum titled DOPC KE 14 eV shows a V-shape not only around the 9th carbon position as pointed out and annotated by the authors but it also around the 5th carbon position (3, 5, 7). However, no C=C exists at the 5th position. It should be clearly pointed out that the "V-shape" is just describing a pattern that can be seen around C=C positions but that not all observable V-shaped peak patterns indicate the existence of C=C.

If not made clear this could be highly confusing and lead to misidentifications.

2. In the abstract the authors claim to identify 96.4% of lipid standards from EAD spectra. Looking at Fig. 2c there is a total of 105 lipid species listed. The major novelty and advantage of EAD over regular CID/HCD-type fragmentation is the resolution of C=C. Given the undiluted row the authors end up at 74 out of 105 lipid species or 70.5% with better than MSL resolution. This is much lower than 96.4% but I find the number still absolutely remarkable. I recommend clarifying the claim of 96% and add more detail to the ratio at various levels of structural resolution as shown in Fig. 2b. The visualization using this dot plot is excellent.

Minor issues:

1. What were the selection/inclusion criteria for those 65 particular lipids used in the main EAD analysis? selection of EV. Fig. 2c shows a completely different number of species. I recommend to unify Fig. 2 in this regard.

2. When presenting the characteristic H-gain peak when identifying PUFAs the authors only show a single example spectrum in Fig. 2d with different KEs. It is hard to follow the argument having this single example. Please add more examples of PUFAs with different omega-positions and at various KEs.

3. In the misidentification spectra of DG 18:1_18:1 and TG 18:1_18:1_18:1 [M+NH4+] the annotated double bond position peak is so low that it is not visible and appears to be missing. Please zoom in so that the peak becomes visible.

4. The authors state that they do not cover the sodium adduct peaks which are covered elsewhere. However, they present the Spingomyelin 18:1(4) with a sodium adduct.

5. There is small inconsistency between "C=C high peak" and "C=C high" peak. Both versions are used in the manuscript. I recommend to unify this.
6. There is not a clear definition of the C=C high peak. Related is Fig. S5d. Typo "shold" -> "should". From my point of view the C=C high peak seems to be there.
7. Lines 592, 593: I am not sure I understand the sentence correctly. What exactly is being predicted and omitted? Please clarify.
8. Line 597: The parameter for the H-gain fragment of 0,05 is never changed.
9. Line 602: just the highest scoring candidate is retained?
10. Lines 604-615: This description is really complicated ("multiples of three etc."). Just simply say omega-6, 3, and 9. This part can be simplified a lot to increase clarity.
11. Fig. 3b contains a lot of colors but I didn't find a description of what those colors represent.
12. Related to the identification of 96.4% as mentioned above. How much was identified in the real biological samples? This would be very interesting so that interested researchers have an idea of what to expect when conducting EAD analyses. Since it is impossible to know exactly what set of species a biological sample contains at least a comparison between EAD and CID identification would showcase the benefits of EAD at its current state.
13. On the MSDIAL tutorial and demonstration page linked in the manuscript there is no EAD tutorial. The section only contains the title of a supposedly intended future section.
14. I find the selected example of highly unsaturated PC34:6/22:6 a little unfortunate and would have preferred a species with different numbers of double bonds on each hydrocarbon moiety.

Overall, the paper is well-written and understandable. I recommend to publish the paper after the raised issues have been thoroughly addressed.

Reviewer #2

(Remarks to the Author)

Takeda et al. use this manuscript to introduce the latest version (v5) of their widely used data-analytics software MS-DIAL. MS-DIAL is widely used in the metabolomics and lipidomics research communities for analysis of complex liquid chromatography-mass spectrometry data obtained from biological samples. A key feature of the new software version is the ability to analyse tandem mass spectra derived from either traditional collision-induced dissociation (and a mainstay of previous versions of MS-DIAL) or electron-activated dissociation which is a modality that has been recently introduced by a single instrument vendor (SCIEX). The authors demonstrate their experimental/computational workflow by describing some unusual glycerophospholipids in ocular lens tissue and purporting to assign both sn-position (i.e., the relative arrangement of fatty acyl chains on the glycerol backbone) and carbon-carbon double bond position (i.e., sites of unsaturation).

The new version of MS-DIAL5 will be of wide interest and utilisation to the user community however there is insufficient evidence that analysis of EAD spectra –be that automated or manual– can provide the claimed structure specific assignments in complex biological extracts. The strategy used in this paper was to build an EAD spectral library using available standards and then to add further standards to the mixture and to re-analyse to see if the original library standards are recapitulated. This is where the claim of 96% comes from. This is an irrelevant confidence metric for a biological extract where there will be numerous isobaric and isomeric contributors to the signal. To consider a simple system. The most abundant membrane glycerophospholipid in most mammalian tissues and cells is the phosphatidylcholine PC 34:1. It is now widely recognised that in most biological systems this species is present as an isomeric mixture of up to 4 isomers, namely, PC 16:0/18:1(9Z), PC 18:1(9Z)/16:0, PC 16:0/18:1(11Z) and PC 18:1(11Z)/16:0. Changes in the relative abundance of these isomer populations has now been well established across tissues and cells and has recently been shown to be a wide spread phenomenon across glycerophospholipid molecular species (doi.org/10.1002/anie.202316793 and [10.1002/anie.201802937](https://doi.org/10.1002/anie.201802937)). As such, to establish the veracity of the key claims of the EAD-MS-DIAL5 pipeline, the authors would need to show how analysis of the spectral and chromatographic evidence can confidently assign these structures and rule in or rule out other isomeric contributors. The authors already acknowledge the challenges of assigning double bond position in lipids with more than one unsaturated chain. It seems highly improbable therefore, given the high noise and complexity of the EAD spectral data, that the workflow can resolve lipid regioisomers. Indeed, the authors should recognise that even the reference standards (including the isotopically labelled versions) they have used contain contributions from more than one sn-isomer as acyl-migration is a feature of wet-chemical synthesis (DOI [10.1194/jlr.M046995](https://doi.org/10.1194/jlr.M046995)). This suggests that the library spectra being generated are hybrids of more than one sn-isomer. In addition, from the raw data provided (e.g., Fig. 3a) the signals being used and data-processed to assign structure do not seem to meet reasonable cut-offs for signal to noise. Overall, the capabilities of the analysis pipeline need to be evaluated against to provide confidence in the molecular assignments and to assess the ability of the method to uniquely assign a single molecular structure in the presence of isomers or isobars. This should include the (i) development of calibration curves of mixed isomer populations against established methods (e.g., enzymatic assays or other benchmarked ion activation modalities such a negative ion CID DOI [10.1194/jlr.M046995](https://doi.org/10.1194/jlr.M046995)) and (ii) spiking in unnatural isomers (e.g., PC 16:0/18:1(8Z)) to establish the thresholds at which

changes in the isomer population can be determined.

The case study of very long chain glycerophospholipids is interesting but the molecules have been described previously in this tissue over 35 years ago by traditional methods (including high energy MIKES, Biochem J.(1988) 253, 645-650). So, these findings are confirmatory rather than novel. The application of imaging to these targets is new and quite interesting but is somewhat independent of the title claims of the manuscript and would be better presented as a separate investigation.

Reviewer #3

(Remarks to the Author)

The manuscript "MS-DIAL 5 multimodal mass spectrometry data mining unveils lipidome complexities" by Takeda et al. addresses an important limitation of structural elucidation of lipids, namely the sn position in phospholipids and triglycerides, and of the double bond in fatty acids and structural details of equal mass lipids. Through electron-activated dissociation (EAD)-based tandem MS, and molecular localization through MS imaging (MSI) the provided algorithm facilitates identification of tissue-specific lipidome. The algorithm was then used to annotate the sn- and double-bond positions of eye-specific phosphatidylcholine molecules containing very-long chain n3 polyunsaturated fatty acids (VLC-PUFAs). Moreover, glycerol 3-phosphate acyltransferase was identified as an enzyme candidate responsible for incorporating n-3 VLC-PUFAs into the sn-1 position of phospholipids.

Several suggestions to further improve the study:

1. The study is technically sound and provides valuable new ways for sn1/sn2 position and double bond analysis in multiple lipid classes. There is, however, limited discussion of new mechanisms or phenomena in the manuscript.

2. The assessments of identification accuracy was done on limited number of lipids mixture. Although the authors discuss that prior sample separation is important for proper identification, this is not always possible due to similar characteristics of several lipid classes. Clear discussion of the limits of analysis would guide the readers to appropriate experimental design.

3. Several prior studies analyzed VLC PUFA and there are several controversies in the field on metabolic pathways affecting VLC PUFA structure and composition in the retina. Please see PMID: 19875612; PMID: 29362226; PMID: 37947776; PMID: 33526677. A discussion of whether this study is in agreement with previously reported VLC PUFA composition and further elucidation of sn1 vs. sn2 position is warranted.

Version 1:

Reviewer comments:

Reviewer #1

(Remarks to the Author)

Thank you for improving the manuscript so far. However, there are still a few questions:

1. [major issues 1] In your answer to Reviewer 1's first comment, you mention that the V-shape peak pattern is used as the minimal criterion for detecting the existence of a double bond at a particular position. This would imply that whenever there is indeed a double bond the V-Shape pattern will be observable. Please confirm that this is ALWAYS the case.

2. [minor issues 11] Thank you for revising Fig. 4b (previously 3b). However, I am still missing a legend explaining the meaning of the colors unless they don't really have a meaning. In case of the latter, please add this information to the figure's caption.

3. [minor issues 12] In the newly created Table S8 is the B&D referring to the original Bligh and Dyer extraction method or to a modified Bligh and Dyer method? Just want to reconfirm.

All other previously raised issues that I do not mention here have been adequately addressed by the authors. I still recommend to publish their work on EAD fragmentation annotation after the last few issues have been addressed.

Reviewer #2

(Remarks to the Author)

Takeda et al. have made significant revisions to their manuscript in response to my initial review and, in so doing, have addressed the two substantial issues I raised. The authors are to be commended for their efforts including a large body of additional experimental work that now clarify the advances and residual limitations of their methodology. Based on this I am supportive of publication subject to consideration of the points below which largely relate to providing additional clarity as to the output of the MSDIALv5 analysis of EAD data.

(1) The additional isomer calibration experiments are excellent and are well summarised in Figure 3 (and elsewhere in supplement). This clarifies a number of key points:

a. that the assignment of double bond position is based on changing peak abundance and not unique fragment ions. As such, the spectral fitting is much more reliant on the training set of spectra and thus the confidence in the assignment of the most abundant double bond isomer is likely heavily reliant on a reference spectrum being available. The authors should make clear in their manuscript that the output is only identifying the most abundant double bond isomer and that the

confidence in this assignment will diminish for more complex isomeric mixtures and/or for unsaturated lipids for which reference spectra are not available.

b. the sn-specific marker ions appear to be more exclusive to individual isomers and so are a more sensitive measure of the presence of multiple isomers. The data from biological samples should therefore be annotated to demonstrate the power of this information. That is it is likely that the annotation can reflect where a mixture of isomers is shown to be present without necessarily making a full molecular lipid assignment. That is, it is biologically significant to be able to indicate that both PC 16:0/18:1 and 18:1/16:0 are present without necessarily being able to assign the double bond position to the sn-position.

(2) Line 308 The statement "...lipids with unique structures, for which the presence of isomers is less expected" is not consistent with the data in the current manuscript or the literature. The manuscript shows that isomers are more often present than absent. It is important to note that the methods outlined in this paper are demonstrated to be effective for identification of the most abundant isomer but do NOT exclude the possibility of other isomers being present. Indeed, even the case study of the VLC-PUFAs should be used to illustrate this. Based on the signal to noise achieved in these spectra and the calibration curves now included the authors would do well to point out explicitly that they cannot rule out the presence of other isomers at up to 10%.

(3) Line 66 "electrospray" not "electron spray"

Reviewer #3

(Remarks to the Author)

The authors fully addressed my comments.

REVIEWER COMMENTS

Reviewer #1

In this manuscript the authors present their work on the software MSDIAL5. MSDIAL has been an important software for the analysis of lipid mass spectra from various experimental setups including different mass spectrometers and recent technology such as TIMS-TOF. In this new iteration number 5 the authors focus on analyzing spectra from electron-activated dissociation (EAD) as provided by the ZenoTOF 7600 mass spectrometer. The authors mainly describe their newly-developed workflow for identifying molecular lipid species with defined carbon-carbon double bond (C=C) positions which is the primary novelty and advantage of EAD fragmentation. This is an important new feature added to MSDIAL 5 to increase the structural resolution in MS-based lipid identification. However, there are a few things that attention.

We appreciate your constructive comments and suggestions to improve our manuscript. We wrote the details more clearly to explain our methodology. Please see our point-by-point responses in blue, and the major changes in the main text highlighted in yellow.

Major issues:

1. In the process of identifying the C=C the authors use a pattern ± 2 Da around a C=C position they call "V-shape". However, this definition is not exhaustive. The sheer existence of a V-shape around a particular position in the spectrum is not indicative of the existence of a C=C at that position. As an example, in Fig. 1b V-Shape the spectrum titled DOPC KE 14 eV shows a V-shape not only around the 9th carbon position as pointed out and annotated by the authors but it also around the 5th carbon position (3, 5, 7). However, no C=C exists at the 5th position. It should be clearly pointed out that the "V-shape" is just describing a pattern that can be seen around C=C positions but that not all observable V-shaped peak patterns indicate the existence of C=C. If not made clear this could be highly confusing and lead to misidentifications.

Thank you. We should clarify that the V-shape pattern is just a marker to consider the possibility of C=C bond. In fact, the most difficult thing in using EAD to determine the double bond positions is the evaluation of the confidence of product ions when considered with the level of noise ions (see lines 825–836 for details of recognizing noise ions in MS-DIAL). Thus, the existence of the increased ions expected in the C=C position is used as the minimum criterion before considering C=C positions. If the criterion is satisfied, the charge remote fragment (CRF) ions related to acyl chains are evaluated between the experimental- and the theoretically generated product ions by using several diagnostic criteria. We added the details in the main text of our revised manuscript (see lines 141–143). The details of diagnostic criteria to rank the candidates in using EAD-MS/MS spectra were further described in **Online method** (see lines 777–824).

2. In the abstract the authors claim to identify 96.4% of lipid standards from EAD spectra. Looking at Fig. 2c there is a total of 105 lipid species listed. The major novelty and advantage of EAD over regular CID/HCD-type fragmentation is the resolution of C=C. Given the undiluted row the authors end up at 74 out of 105 lipid species or 70.5% with better than MSL resolution. This is much lower than 96.4% but I find the number still absolutely remarkable. I recommend clarifying the claim of 96% and add more detail to the ratio at various levels of structural resolution as shown in Fig. 2b. The visualization using this dot plot is excellent.

Thank you. In this revision, we reassessed the results of MS-DIAL 5 validation, as shown in **Figure 2c**, following the definition of structural description in **Figure 2b**. These results are included in the newly created **Supplementary Table 6**. When integrating results from all concentration ranges of the dilution series data, 62.1% of the correctly annotated peaks as species level (SL) or molecular species level (MSL) could be characterized with *sn*-, OH-, and/or C=C positional information where the resolution of lipid structure description exceeded that of conventional CID-MS/MS-based lipidomics, describing lipids with the molecular species level (MSL) or species level (SL). We evaluated the results of MS-DIAL 5 using samples with concentrations labeled as "x20 (20-times diluted)" for LightSPLASH, and "x2 (2-times diluted)" for Ultimate SPLASH and the in-house standard mixture, as shown in **Figure 2c**. This range corresponds to samples adjusted to approximately 1 μ M or more. Since 1 μ L of 1 μ M sample contains a molecule of 1,000 femtomoles, this range evaluates concentrations where essential fragment ions for *sn*- and C=C position determinations are observed as shown in **Figure 2a**. The evaluation within this range showed that MS-DIAL 5 could achieve in-depth structure elucidations for 78.0% of the correctly annotated peaks as MSL or SL descriptions. However, at high concentration ranges, especially for neutral lipids such as triacylglycerol (TG) and diacylglycerol (DG), there was an increased possibility of misannotation, as discussed in the main text following **Supplementary Figure 6**. We wrote these results in the abstract and the main text sections accordingly (see lines 52–54 and 174–179).

Minor issues:

1. *What were the selection/inclusion criteria for those 65 particular lipids used in the main EAD analysis? selection of EV. Fig. 2c shows a completely different number of species. I recommend to unify Fig. 2 in this regard.*

Practically, the lipid authentic standards stored in our laboratory or collaborators were used to formulate the EAD-fragmentation patterns implemented in the MS-DIAL 5 program. First, we tried to prepare a variety of lipid subclasses, resulting in the confirmation of EAD-MS/MS spectral patterns for 43 unique lipid subclasses. Furthermore, we prepared the acyl chain isomers, including saturated, monosaturated, and polyunsaturated fatty acids of phosphatidylcholine (PC) and phosphatidylethanolamine (PE), to elucidate the mechanism of charge remote fragmentation (CRF) on lipid acyl chains. The MS/MS spectra of some oxidized lipids and branched chains containing lipids were acquired, although the annotation is not supported yet in MS-DIAL due to less spectral information of authentic standards to be formulated. It was beyond the scope of this work.

We summarize the workflow of this study. First, we implemented the algorithm for lipid annotations by confirming the EAD-MS/MS spectra of 65 standard compounds. Second, the program was validated by the dilution series data of LightSPLASH containing 13 lipid standards, 11 lipids not included among the 65 standard compounds. The lipids were simply dissolved in methanol. Finally, the program was validated based on the dilution series data of 91 lipid standards, which were added to the biological extract of the ^{13}C -labeled plant as a sample matrix. Because the matrix background between the 13 lipids mixture and the 91 lipids mixture differed, we evaluated the validation results separately. To explain the details more clearly, we modified **Figure 2c** and added the relevant explanation in the main text (see lines 155–160).

2. *When presenting the characteristic H-gain peak when identifying PUFAs the authors only show a single example spectrum in Fig. 2d with different KEs. It is hard to follow the argument having this single example. Please add more examples of PUFAs with different omega-positions and at various KEs.*

We created new **Supplementary Figure 3**, which includes the same MS/MS spectral descriptions as described in **Figure 1d**, for PC 18:3(9,12,15)/18:3(9,12,15), PC 22:6(4,7,10,13,16,19)/22:6(4,7,10,13,16,19), PC O-16:0/20:4(5,8,11,14), PE 18:0/20:4(5,8,11,14), PE 22:6(4,7,10,13,16,19)/22:6(4,7,10,13,16,19), and PG 22:6(4,7,10,13,16,19)/22:6(4,7,10,13,16,19). The original data resources are available with the indexes of TUAT00019, TUAT00021, TUAT00023, TUAT00025, TUAT00028, and TUAT00032, respectively, in the RIKEN DROPMet repository (ID: DM0054). On examination of the EAD-MS/MS spectra derived from these seven lipid molecules, a significant increase in ion intensity from H-gain fragments was confirmed for all molecules when using kinetic energies (KE) of 14 eV and 18 eV. Moreover, 14 eV showed higher sensitivity than 10 eV. These results show the high scalability of our discovery related to the H-gain fragment behavior. We appreciate your suggestion, and the revised sentences were added to the main text (see lines 105–108).

3. In the misidentification spectra of DG 18:1_18:1 and TG 18:1_18:1_18:1 [M+NH₄⁺] the annotated double bond position peak is so low that it is not visible and appears to be missing. Please zoom in so that the peak becomes visible.

We added the figures for the visibility of the double-bond peaks in **Supplementary Figures 2 and 4**.

4. The authors state that they do not cover the sodium adduct peaks which are covered elsewhere. However, they present the Spingomyelin 18:1(4) with a sodium adduct.

Thank you for your comment. We realized that we need to provide more details on the current status of LC-MS/MS and MS-DIAL 5. We obtained MS/MS spectra of sodium adduct ions for authentic standards. For phospholipids and glycerolipids, the EAD technique generates clearer ions related to *sn*-positions in sodium adduct ion forms, highlighting the importance of sodium adducts for in-depth structural lipidomics. MS-DIAL 5 supports the annotation of MS/MS spectra derived from sodium adduct ions. However, in this study we did not evaluate how the system of LC-MS/MS and MS-DIAL can be applied to MS/MS spectra of sodium adducts. In particular, we have not evaluated false and true hits as performed in **Figure 2c**. To increase the ion intensity of sodium adducts, the post-column method flowing sodium acetate solution is effective in increasing the sensitivity of sodium adduct ion forms (e.g. as described in Kato S. et al. *npj Sci Food* 2, 1 (2018); doi.org/10.1038/s41538-017-0009-x). However, this method reduces the relative sensitivity of [M+H]⁺ and [M+NH₄]⁺ ions, and the amount of sodium ions generated is often less than expected. This is due to low mixing efficiency at the sodium acetate merging point and low ionization efficiency in electrospray ionization using conventional LC-MS/MS systems installing semi-microcolumns. Therefore, we recently installed a micro-flow LC system and optimized the materials, flow rates, and solvent conditions to efficiently acquire the sodium adduct ions using the post-column method, resulting in the acquisition of high-quality EAD-MS/MS spectra. In the near future, we plan to publish comprehensive structural lipidomics data focusing on sodium adducts using micro-flow LC and MS-DIAL 5 as a follow-up to this study.

5. There is small inconsistency between "C=C high peak" and "C=C high" peak. Both versions are used in the manuscript. I recommend to unify this.

We used the term “C=C high” peak in the revised manuscript.

6. *There is not a clear definition of the C=C high peak. Related is Fig. S5d. Typo "shold" -> "should". From my point of view the C=C high peak seems to be there.*

We added the zoomed-in figure to **Supplementary Figure S6d**. We hope that the readers note the absence of the "C=C high" peak in the experimental spectrum. The fragment ion of m/z 497.2482 was not considered as a "C=C high" peak because there was a 0.2 Da difference from the theoretical m/z value, while the mass accuracy of our data was <0.01 Da.

7. *Lines 592, 593: I am not sure I understand the sentence correctly. What exactly is being predicted and omitted? Please clarify.*

We modified the sentences as follows:

If more than five "C=C high" peaks were predicted, indicating the presence of three or more double bonds within the acyl chain, the algorithm was designed to allow annotation of double bond positions even if one of the "C=C high" peaks was missing. For example, if there were three double bonds, six "C=C high" peaks were expected (or seven if the acyl chain contained a methylene-interrupted C=C sequence, such as linolenic acid, producing "C=C PUFA high" peak). The program would attempt to determine the double bond positions even if one of these six peaks was not detected; that is, if two or more of the six peaks were missing, the program would not proceed with double bond position annotation. (see lines 782–788)

8. *Line 597: The parameter for the H-gain fragment of 0,05 is never changed.*

We added the following sentence:

The factor of the H-gain fragment was changed to 4.0 for the "C=C PUFA high" peak. (see line 795-796)

9. *Line 602: just the highest scoring candidate is retained?*

The MS-DIAL 5 program internally retains other candidates exceeding the diagnostic criteria. We created the environment to confirm at least up to the top three candidates in the EAD-based annotation in the GUI. All candidates can be browsed using the compound search function as described below.

10. Lines 604-615: This description is really complicated ("multiples of three etc."). Just simply say omega-6, 3, and 9. This part can be simplified a lot to increase clarity.

Thank you. We modified the relevant sentences as follows (see line 837–847):

The structural diversity of the C=C positional isomers generated in MS-DIAL is inherently limited, and the configuration is seemingly optimized for mammalian cells. For monounsaturated fatty acids (MUFAs) with *O*-acyl and *N*-acyl chains, the potential C=C positions were derived from those listed in the LIPID MAPS Structure Drawing Tool for glycerophospholipid structures. Positions defined as multiples of three from the omega terminus, i.e., omega 3, 6, and 9, were included as potential sites. The C=C positions of sphingoid bases were obtained using the candidate list from the LIPID MAPS tool for sphingolipids, including delta 4, 6, 8, and 14, as the reference. Of these, the candidates of delta 6 were excluded because the C=C position was unusual in mammalian cells. Furthermore, only the delta 4 position is considered when the sphingoid base contains one double bond. For PUFAs not included in LIPID MAPS, candidate structures of omega 3, 6, and 9 fatty acids, including VLC-PUFA, were generated.

11. *Fig. 3b contains a lot of colors but I didn't find a description of what those colors represent.*

We simplified the figure so that users understand all the contents of Figure 4b (which was Figure 3b originally), as shown below.

12. Related to the identification of 96.4% as mentioned above. How much was identified in the real biological samples? This would be very interesting so that interested researchers have an idea of what to expect when conducting EAD analyses. Since it is impossible to know exactly what set of species a biological sample contains at least a comparison between EAD and CID identification would showcase the benefits of EAD at it's current state.

We summarized the annotation results in the newly created Supplementary Table 8.

Table S8. Annotation results using CID-MS/MS and EAD-MS/MS for mouse eye lipidome

	SN_DB	OH_DB	SN	OH	DB	MSL	SL	Total
POS-EAD-MS/MS (SPE)	16	4	30	10	2	56	0	118
POS-EAD-MS/MS (B&D)	34	8	88	53	42	341	0	566
POS-CID-MS/MS (B&D)	0	0	0	0	0	681	90	771
NEG-CID-MS/MS (B&D)	0	0	0	0	0	567	26	593

SPE means solid phase extraction

B&D means bligh and dyer lipid extraction method

The definitions of SN_DB, OH_DB, SN, OH, DB, MSL, SL were the same as used in Figure 2.

The above results were based on LC-MS/MS data analyzing four biological replicates.

The species level (SL) annotation was not evaluated in the EAD-MS/MS mode. Nevertheless, the summary table indicates the advantages and the disadvantages of the EAD-MS/MS strategy. The annotation coverage in EAD-MS/MS with the Bligh and Dyer (B&D) lipid extraction method was 73% (566/771) lower than that of positive ion mode (POS)-CID-MS/MS data, indicating that the sensitivity of product ion peaks for annotation is higher in CID-MS/MS than that in EAD-MS/MS. The LC-EAD-MS/MS data, with the support of MS-DIAL 5, achieved in-depth structure elucidations for 225 lipids; in fact, the number reached 250 when the result from the solid phase extraction (SPE) method was integrated where the duplicates were excluded. MS-DIAL 5 provided in-depth structural descriptions for approximately 40% (225/566) chromatographic peaks in LC-EAD-MS/MS data. We clarified the pros and cons in the main text. (see line 309–315)

13. On the MSDIAL tutorial and demonstration page linked in the manuscript there is no EAD tutorial. The section only contains the title of a supposedly intended future section.

We apologize for this oversight. We uploaded the tutorial document and video on the website.

14. I find the selected example of highly unsaturated PC34:6/22:6 a little unfortunate and would have preferred a species with different numbers of double bonds on each hydrocarbon moiety.

A new **Supplementary Figure 9** was created, containing the product ion annotations for PC 16:0/22:6(4,7,10,13,16,19), PC 18:0/22:6(4,7,10,13,16,19), PC 16:0/20:4(5,8,11,14), and PC 34:5(19,22,25,28,31)/22:6(4,7,10,13,16,19) detected in mouse eye samples. These lipid molecules contain polyunsaturated fatty acids in their acyl chains and are among the five most abundant diacyl phosphatidylcholines with different molecular species. In the lipidomics data obtained via the Bligh and Dyer method, PC 34:6(16,19,22,25,28,31)/22:6(4,7,10,13,16,19), as illustrated in the primary figure (**Figure 4a**), was the second most abundant peak among those meeting the above criteria. In fact, PC 16:0/22:6(4,7,10,13,16,19) is the most abundant molecule satisfying these criteria. The EAD-MS/MS spectra for each lipid species have been interpreted. Furthermore, the double bond positions obtained from these results are consistent with those obtained with an orthogonal technique using oxygen attachment dissociation (OAD)-MS/MS (<https://www.nature.com/articles/s42004-022-00778-1>). The results are described in the main text (see lines 315–321).

Overall, the paper is well-written and understandable. I recommend to publish the paper after the raised issues have been thoroughly addressed.

Thank you for providing constructive comments and suggestions which have helped us improve our paper.

Reviewer #2 (Remarks to the Author):

Takeda et al. use this manuscript to introduce the latest version (v5) of their widely used data-analytics software MS-DIAL. MS-DIAL is widely used in the metabolomics and lipidomics research communities for analysis of complex liquid chromatography-mass spectrometry data obtained from biological samples. A key feature of the new software version is the ability to analyse tandem mass spectra derived from either traditional collision-induced dissociation (and a mainstay of previous versions of MS-DIAL) or electron-activated dissociation which is a modality that has been recently introduced by a single instrument vendor (SCIEX). The authors demonstrate their experimental/computational workflow by describing some unusual glycerophospholipids in ocular lens tissue and purporting to assign both sn-position (i.e., the relative arrangement of fatty acyl chains on the glycerol backbone) and carbon-carbon double bond position (i.e., sites of unsaturation).

Thank you very much for your constructive suggestions, which have helped us improve our manuscript. We performed several additional experiments according to your comments. Please see our point-by-point responses in blue, and the major changes in the main text highlighted in yellow.

The new version of MS-DIAL5 will be of wide interest and utilisation to the user community however there is insufficient evidence that analysis of EAD spectra –be that automated or manual– can provide the claimed structure specific assignments in complex biological extracts. The strategy used in this paper was to build an EAD spectral library using available standards and then to add further standards to the mixture and to re-analyse to see if the original library standards are recapitulated. This is where the claim of 96% comes from. This is an irrelevant confidence metric for a biological extract where there will be numerous isobaric and isomeric contributors to the signal. To consider a simple system. The most abundant membrane glycerophospholipid in most mammalian tissues and cells is the phosphatidylcholine PC 34:1. It is now widely recognised that in most biological systems this species is present as an isomeric mixture of up to 4 isomers, namely, PC 16:0/18:1(9Z), PC 18:1(9Z)/16:0, PC 16:0/18:1(11Z) and PC 18:1(11Z)/16:0. Changes in the relative abundance of these isomer populations has now been well established across tissues and cells and has recently been shown to be a wide spread phenomenon across glycerophospholipid molecular species (doi.org/10.1002/anie.202316793 and [10.1002/anie.201802937](https://doi.org/10.1002/anie.201802937)). As such, to establish the veracity of the key claims of the EAD-MS-DIAL5 pipeline, the authors would need to show how analysis of the spectral and chromatographic evidence can confidently assign these structures and rule in or rule out other isomeric contributors. The authors already acknowledge the challenges of assigning double bond position in lipids with more than one unsaturated chain. It seems highly improbable therefore, given the high noise and complexity of the EAD spectral data, that the workflow can resolve lipid regioisomers. Indeed, the authors should recognise that even the reference standards (including the isotopically labelled versions) they have used contain contributions from more than one sn-isomer as acyl-migration is a feature of wet-chemical synthesis (DOI 10.1194/jlr.M046995). This suggests that the library spectra being generated are hybrids of more than one sn-isomer. In addition, from the raw data provided (e.g., Fig. 3a) the signals being used and data-processed to assign structure do not seem to meet reasonable cut-offs for signal to noise. Overall, the capabilities of the analysis pipeline need to be evaluated against to provide confidence in the molecular assignments and to assess the ability of the method to uniquely assign a single molecular structure in the presence of isomers or isobars. This should include the (i) development of calibration curves of mixed isomer populations against established methods (e.g., enzymatic assays or other benchmarked ion activation modalities such a negative ion CID DOI 10.1194/jlr.M046995) and (ii) spiking in unnatural isomers (e.g., PC

16:0/18:1(8Z)) to establish the thresholds at which changes in the isomer population can be determined.

In this round of review and resubmission, we conducted additional experiments according to your suggestions, whose results were described in newly created **Figure 3**, **Supplementary Figure 7**, and **Supplementary Table 7**. The following IsoPure standards were purchased: PC 16:0/18:1(9Z), PC 16:0/18:1(11Z), and PC 18:1(9Z)/16:0. The purity of these IsoPure standards was confirmed by Avanti Lipids. By C13 NMR, IsoPure PC was confirmed to show less than 5% acyl migration. These products are the most effective lipid authentic standards to evaluate the MS-DIAL 5 program for EAD-MS/MS spectral data. Before seeing the following response letter, please see the main text (lines 215-304) which contains the results and discussions for these additional experiments. Briefly, the MS-DIAL 5 program was applied to (A) the mixture of these authentic standards with various concentrations, (B) mouse brain, and (C) NIST SRM 1950 human plasma, for which lipid elucidations have been performed by the other fragmentation techniques so far (**Figure 3** as attached). Consequently, **MS-DIAL 5 characterized the most abundant lipid isomer as the top hit candidate, indicating the high scalability of MS-DIAL 5 for annotation in co-eluted chromatogram peaks**. Importantly, we validated the results of C=C positional isomer ratios in biological samples using an oxygen attachment dissociation (OAD)-based MS/MS technique as an orthogonal method.

First, the retention times of these lipids were confirmed using the 25-minute LC condition used in this study (newly created **Supplementary Figure 7a**). The results indicated that the retention times of these lipids are mostly equivalent and not differentiated, suggesting that the product ions obtained from biological samples will exhibit a mixed spectrum of these isomers. To increase the throughput of additional experiments, we employed a liquid chromatography condition where the injection-to-injection time is approximately 9 minutes. This methodology is based on our previous research (K. Tokiyoshi et al. *Analytical Chemistry* 96 (3), 991-996, 2024). In addition, the fragmentation patterns of the lipid isomers were investigated (**Supplementary Figure 7b**). The purpose of the additional experiments is to generate a dataset for evaluating MS-DIAL 5 in the context of a real biological sample, with a particular focus on the profiles of the four well-characterized isomers PC 16:0/18:1(9Z), PC 16:0/18:1(11Z), PC 18:1(9Z)/16:0, and 18:1(11Z)/16:0. However, as multistage fragmentations, such as MS3, are not available in SCIEX EAD-MS/MS, the ratio of delta 9 and delta 11 C=C positional isomers is derived from the summed value of the *sn1*- and *sn2* positional isomers. The same is true for the ratio of *sn1*- and *sn2* positional isomers, whereby the result represents the summed value of the abundances of delta 9 and delta 11 C=C positional isomers.

Calibration curves were constructed for each of the authentic standards to gain insight into the linear range of MS1 and product ion peaks (**Supplementary Figure 7c**). The product ion spectrum was obtained using targeted MS/MS mode, whereby the defined precursor *m/z* value was isolated. The dilution series was prepared by adjusting the concentration from 5 nM to 5 μ M where the injection volume was set to 1 μ L; the same injection volume was used for subsequent experiments. Three technical replicates (N=3) were prepared. By using the peak height of the extracted ion chromatogram (EIC) of the precursor ion *m/z* in the MS1 data, we obtained a linear relationship with an R-square value exceeding 0.99 within the concentration range of 5 nM to 5 μ M. In contrast, the C=C high peaks to evaluate C=C positions were often not detected around 50 nM, suggesting that the limit of detection in our LC-MS/MS system is 50 fmol on-column injection, and the same was true for *sn*-position specific diagnostic ion (**Supplementary Figures 7c**). Nevertheless, in any range exceeding 50 nM, for instance, in the case of PC 16:0/18:1(9Z), the V-shaped pattern in the product ion spectrum at C7-C9-C11, was sustained. Moreover, while the peak intensity of *sn1*-16:0 increased in a concentration-dependent manner, the product ion peak intensity derived from *sn1*-18:1(9) was either undetected or identified as a “barcode ion” across all concentration ranges. The details to recognize the noise ions were described in Online method (see **lines 825-836**).

Calibration curves for diagnostic ions used to determine the double bonds (C=C high peaks) and *sn*-positional isomers demonstrated that the linear range is from 50–100 nM until 500 nM–1000 nM. Based on the calibration curve data, the standard mixtures of varying concentrations were prepared for the pairs of PC 16:0/18:1(9Z) and PC 16:0/18:1(11Z), as well as PC 16:0/18:1(9Z) and PC 18:1(9Z)/16:0. The final concentrations of the pair of (A) PC 16:0/18:1(9Z) and (B) PC 16:0/18:1(11Z) were adjusted to A:B=1000 nM:100 nM, A:B=750 nM:250 nM, A:B=500 nM:500 nM, A:B=250 nM:750 nM, and A:B=100 nM:1000 nM. Calibration curves were constructed with the ratio of diagnostic ion peak heights derived from isomers on the Y-axis and the concentration ratio on the X-axis (**Supplementary Figures 7d**). The result showed a high correlation (R-square = 0.938) between the concentration ratio of *sn*-positional isomers (i.e., *sn1*-16:0 vs *sn1*-18:1) and the intensity ratio. The correlation between the peak intensity ratio of diagnostic ions indicating C=C positions and the concentration ratio was relatively low, even for H-loss fragment ions (R-square = 0.87), which have higher reproducibility than that of C=C high radical fragment ions. This is because the same *m/z* values of the product ion peaks from charge remote fragmentation related to acyl chains are commonly observed among isomers. Therefore, constructing such calibration curves is essential when quantifying C=C positional isomers using EAD-MS/MS in biological samples. To obtain reliable annotations, it is necessary to improve chromatographic separation or confirm

with an orthogonal method to distinguish isomers. According to our investigations, the qualitative and quantitative reliability of co-eluting double bond positional isomers in EAD-MS/MS is inferior to that of other methods, such as oxygen attachment dissociation (OAD), ozone-induced dissociation (OzID), and Paterno-Büchi reaction-based techniques. Nevertheless, there is an advantage in EAD-MS/MS for the deep lipid structure elucidation in the MS2 spectral dimension without any derivatization method. Furthermore, annotation of the most abundant co-eluted isomer can be facilitated with the support of MS-DIAL, as described below. One example showing the advantage is the characterization of VLC-PUFA PC in the mouse retina targeted in this study.

The LC-MS data of these mixture calibration curves are used to evaluate MS-DIAL (newly created **Figure 3**). The result indicated that MS-DIAL annotates the lipid isomer with higher concentration in the mixture as the top hits. The only exception was the case of PC 16:0/18:1(9) 750 nM vs. PC 16:0/18:1(11) 250 nM, where the lower concentration molecule PC 16:0/18:1(11) was characterized as the top hit in one of three replicates. The results indicated that the *sn*-position was accurately characterized by the high-concentration molecules. The accuracy for annotating the most abundant lipid molecule was 91.7% for the C=C position and 100% for *sn*-position characterizations.

Finally, we generated the LC-EAD-MS/MS data of mouse brain and NIST SRM 1950 plasma, which have been well-characterized for the profiles of PC 16:0_18:1 isomers. Furthermore, an evaluation using the same mouse brain lipid extract where an authentic standard PC 16:0/18:1(8), which is not present in animal tissues was spiked for emulating contamination of unexpected molecules, was also performed. The results were described as follows in the main text.

We employed our LC-EAD-MS/MS and MS-DIAL 5 system to characterize PC 16:0_18:1 isomers in the mouse brain and NIST SRM 1950 plasma, as their abundances have been previously characterized. The MS-DIAL 5 program characterized PC 16:0/18:1(9) as the most probable lipid isomer in both mouse brain and human plasma (**Figure 3c and Supplementary Table 7**), which agrees with the previous studies. We added a feature in MS-DIAL 5 to show other possible peak annotations, which could potentially reveal co-eluting lipids. In the brain, PC 18:1(9)/16:0 was annotated as the second potential candidate with a high confidence score, as evidenced by the detection of both diagnostic ions for the *sn*- and C=C positions (see the detail in **Online method: lines 777-824**). PC 16:0/18:1(11) and PC 18:1(11)/16:0 were also assigned as the third and fourth highest-scoring candidates, respectively. On the other hand, PC 16:0/18:1(9) is the only candidate with a high confidence score in human plasma. Data from previous studies suggest that *sn*1-18:1 isomers, as well as isomers containing 18:1 (δ -11) are more abundant in brain than in plasma. Thus, MS-DIAL5 correctly identified the expected co-eluting lipids as candidates. Moreover, the abundance ratio of *sn*1-16:0 and *sn*1-18:1, along with the ratio of δ -9 and δ -11 isomers, could be estimated utilizing calibration curves made from the above mixtures of synthetic lipids (**Supplementary Figure 7d**). The isomer with *sn*1-16:0 and *sn*2-18:1 (δ -9) showed high abundance in both samples, while other isomers were more abundant in brain than in plasma (**Supplementary Figure 7e**), consistently with previous studies. To further validate our estimation of the abundances of δ -9 and δ -11 isomers, an orthogonal method using OAD-MS/MS was employed with the same samples (**Supplementary Figure 7f, 7h, and 7i**). The estimated ratio of δ -9 and δ -11 isomers gave consistent values between OAD-MS/MS and EAD-MS/MS (**Supplementary Figure 7e**). Therefore, MS-DIAL5 correctly identified abundant co-eluting lipids, with quantitative estimation being possible using synthetic standard. Further evaluation of MS-DIAL 5 was conducted using the same mouse brain lipid extract. An authentic standard, PC 16:0/18:1(8), not present in animal tissues and not included in the search space of the MS-DIAL program, was spiked to emulate contamination by unexpected molecules. MS-DIAL 5 annotated PC 16:0/18:1(9) as the top candidate in the samples spiked with 100

nM and 500 nM of PC 16:0/18:1(8) in which the endogenous PC 16:0_18:1 was adjusted to 1000 nM in LC-MS vials (**Figure 3d and Supplementary Table 7**). In the sample spiked with 1000 nM, one of the three technical replicates was annotated as PC 16:0/18:1(9), while the other two were characterized as PC 16:0/18:1(11) and PC 16:0/18:1(13). From all these data, we concluded that the current LC-EAD-MS/MS and MS-DIAL 5 technique can at least accurately annotate the most abundant lipids for the co-eluted peaks, as validated for monounsaturated fatty acid-containing PC in this study. The accuracy of spectral annotation can be reduced when isomers co-elute, particularly in the case of C=C positional isomers, highlighting the necessity for further development in sample preparation and chromatography techniques to fully leverage the potential of EAD-MS/MS. Nevertheless, our current data already suggest that co-eluting lipid abundance can be estimated if synthetic standards are available. Moreover, our results clearly indicated that there is an advantage in the EAD-MS/MS for the characterization of *sn1/sn2* positional isomers because the *sn1*-specific fragment ions are less contaminated from the charge remote fragment ions related to acyl chains.

Base on these additional experiments and data analyses, we could efficiently evaluate MS-DIAL 5 for the case of co-eluting lipids. Furthermore, as discussed in **lines 818-824**, we could estimate the threshold for the annotation of both *sn*- and C=C positional isomers of PC. We believe that our manuscript was substantially improved by your suggestions. Thank you very much.

The case study of very long chain glycerophospholipids is interesting but the molecules have been described previously in this tissue over 35 years ago by traditional methods (including high energy MIKES, Biochem J.(1988) 253, 645-650). So, these findings are confirmatory rather than novel. The application of imaging to these targets is new and quite interesting but is somewhat independent of the title claims of the manuscript and would be better presented as a separate investigation.

We understand your opinion. Nevertheless, there are several reasons for using the retina lipidome data in this manuscript. First, the aim was to demonstrate a novel bioengineering technique for new discoveries in life sciences. Although you mentioned that our finding is instead a confirmation, we believe that this is the first study showing the VLC-PUFA PC structure as an intact form. In this revision, we performed an additional experiment where the incorporation of VLC-PUFA into the *sn1* position of PC in HeLa cells with the supplementation of VLC-PUFA was confirmed by our methodology. This is a novel aspect that has never been reported elsewhere (see the details in the response to the first comment of reviewer #3). While we cannot rule out the possibility that retina-specific incorporation pathways for VLC-PUFAs exist, our platform identified that *sn1* specificity is also achieved in cultured cells, leading to the discovery of GPATs as enzymes sufficient for VLC-PUFA incorporation into glycerophospholipids. Thus, we believe that the discovery of GPAT recognition system can be reported in this paper.

Reviewer #3 (Remarks to the Author):

The manuscript “MS-DIAL 5 multimodal mass spectrometry data mining unveils lipidome complexities” by Takeda et al. addresses an important limitation of structural elucidation of lipids, namely the sn position in phospholipids and triglycerides, and of the double bond in fatty acids and structural details of equal mass lipids. Through electron-activated dissociation (EAD)-based tandem MS, and molecular localization through MS imaging (MSI) the provided algorithm facilitates identification of tissue-specific lipidome. The algorithm was then used to annotate the sn- and double-bond positions of eye-specific phosphatidylcholine molecules containing very-long chain n3 polyunsaturated fatty acids (VLC-PUFAs). Moreover, glycerol 3-phosphate acyltransferase was identified as an enzyme candidate responsible for incorporating n-3 VLC-PUFAs into the sn-1 position of phospholipids.

Thank you so much for your valuable comments and constructive suggestions which helped us improve our manuscript. Please see our point-by-point responses described below.

Several suggestions to further improve the study:

1. The study is technically sound and provides valuable new ways for sn1/sn2 position and double bond analysis in multiplelipid classes. There is, however, limited discussion of new mechanisms or phenomena in the manuscript.

We interpret your comment as a suggestion to engage in a more detailed explanation of the novel aspects and emerging mechanisms in the field of lipid biology in mouse retina. If the comment pertains to the mechanisms of mass fragmentation, we offer our sincerest apologies. However, our manuscript does not propose new mechanisms of mass fragmentation; rather, it focuses on developing the MS-DIAL 5 program to efficiently decode the complex spectra from EAD-MS/MS. In fact, the finding of a substantial increase in H-gain fragment ion abundance at 14 eV to 18 eV kinetic energy conditions, which facilitates the structure elucidation of polyunsaturated fatty acids (PUFAs), is a novel discovery in the MS-based analytical chemistry; the mechanism has already been discussed in the main text.

By applying our method to the lipid profile of the mouse eye, we investigated the phenomenon of very long-chain polyunsaturated fatty acids (VLC-PUFAs) as being enriched in the *sn1* position in retinal tissue. Our hypothesis was that GPAT1 may be partially responsible for incorporating VLC-PUFA into the *sn1* position of phospholipids. The biosynthesis and function of VLC-PUFAs have been studied with the investigations of very long chain fatty acid elongase 4 (ELOVL4). Our study investigates the subsequent process by which VLC-PUFAs, produced by ELOVL4, are incorporated into the *sn1* position of phosphatidylcholine (PC). As demonstrated in our study, the ACSL6 knockout mice, which preferentially utilize DHA as a substrate, did not exhibit a reduction in VLC-PUFA PCs. Moreover, our new experimental data in this revision for the structure elucidation of VLC-PUFA PCs in HeLa cells supplemented with VLC-PUFA indicated that incorporating VLC-PUFA into the *sn1* position of PC in HeLa cells was valid. Given that the substrate recognition sites for GPAT1, ELOVL4, and acyl CoA synthetases (ACSLs) are in the cytoplasm, it is feasible that these three enzyme's substrates and products can be shared within the cytoplasm. Based on these facts, VLC-PUFA produced by ELOVL4 may be converted into VLC-PUFA CoA by an ACSL other than ACSL6. Our study demonstrated that the VLC-PUFA-CoA is incorporated into the *sn1* position of PC in HeLa cells, which supports the hypothesis that VLC-PUFA is metabolized to LPA *sn1*-VLC-PUFA by GPAT. While we cannot rule out the possibility that retina-specific incorporation pathways for VLC-PUFAs exist, our platform identified that the *sn1*

specificity is also achieved in cultured cells, leading to the discovery of GPATs as enzymes sufficient for VLC-PUFA incorporation into glycerophospholipids. Further investigations for biochemistry related to ACSL1~6, ELOVL4, and the isozymes of GPAT in the VLC-PUFA PC biosynthesis are needed. The above discussions have been added to the revised manuscript (see lines 361–376).

2. The assessments of identification accuracy was done on limited number of lipids mixture. Although the authors discuss that prior sample separation is important for proper identification, this is not always possible due to similar characteristics of several lipid classes. Clear discussion of the limits of analysis would guide the readers to appropriate experimental design.

Through this study, it is evident that methodological advancements are required in sample preparation and liquid chromatography to facilitate the lipid structure elucidation with the EAD-MS/MS technique at the lipidomics scale. The results of the additional experiments performed in this revision process indicated that a sample concentration of approximately 200 nM to 1000 nM is required to achieve reliable structure elucidation with EAD-MS/MS (see the detail in the response to reviewer #2). Given that oxylipins in human plasma are approximately 50 pM and the most abundant lipid, cholesterol, is around 5 mM, the concentration range of lipids spans eight orders of magnitude. To obtain high-quality EAD-MS/MS spectra for low abundant lipids, lipid enrichment is essential. The purification of lipids of interest is also an emerging need to avoid contamination caused by excessive sample injection into mass spectrometry.

In recent studies, we achieved the successful fractionation of phospholipids, neutral lipids, and fatty acids/glycolipids using solid-phase extraction columns with titanium and zirconium, through the optimization of solvent conditions. This method is particularly useful for the enrichment of lipids, such as fatty acids and glycolipids, which can be separated from phospholipids and neutral lipids like triacylglycerols (<https://www.biorxiv.org/content/10.1101/2023.10.15.562428v2.abstract>). In fact, a recent report indicates that titanium-based beads can be employed for the easy purification of glycolipids, facilitating comprehensive structure elucidation of sphingoglycolipids through Paterno-Büchi reactions (<https://www.nature.com/articles/s41467-024-50014-8>). For isolating specific phospholipid species, such as VLC-PUFA PC, additional considerations are necessary. Due to their elevated hydrophobicity relative to other phospholipids containing saturated and monounsaturated fatty acids, VLC-PUFA PCs could be fractionated using C18 reverse-phase solid-phase columns. However, classical thin-layer chromatography methods may also be a viable option for fractionation for lipid classes with typically low abundance, such as PS, PI, and PG.

Given that EAD-MS/MS causes bond cleavages at all bond sites in principle, distinguishing double bond positions becomes particularly challenging when isomers co-elute. This has been demonstrated using standard compounds of three lipid molecules. The separation of PC 16:0/18:1(9), PC 16:0/18:1(11), and PC 18:1(9)/16:0 is challenging under typical chromatographic conditions (see the responses to the comments of Reviewer #2). Accordingly, the current reverse-phase LC methods are inadequate for fully utilizing the capabilities of EAD-MS/MS, underscoring the necessity for future developments in liquid chromatography.

In conclusion, informatics tools like MS-DIAL 5 have significantly lowered the barriers to data analysis, making lipidomics-scale profiling with EAD-MS/MS theoretically possible. However, it is inadvisable to rely solely on the capabilities of informatics tools, and further development in analytical chemistry is necessary to obtain pure spectra data as much as possible. The above discussions have been added to the revised manuscript (see lines 215–304).

3. Several prior studies analyzed VLC PUFA and there are several controversies in the field on metabolic pathways affecting VLC PUFA structure and composition in the retina. Please see PMID: 19875612; PMID: 29362226; PMID: 37947776; PMID: 33526677. A discussion of whether this study is in agreement with previously reported VLC PUFA composition and further elucidation of *sn1* vs. *sn2* position is warranted.

Thank you so much for this comment. We carefully reviewed the four papers and several other studies on the metabolic pathways affecting VLC-PUFA. In the first two references (PMID 19875612 and 29362226), it has been shown that VLC-PUFAs are synthesized by the elongase ELOVL4 and are incorporated into various lipids, such as phosphatidylcholine and ceramide. We believe the discrepancy in VLC-PUFA incorporation can be attributed to the differing capabilities of the acyltransferases and ceramide synthases expressed in distinct cells in utilizing VLC-PUFA-CoA. The references, PMID: 37947776 and 33526677, describe the chemical synthesis of VLC-PUFAs for bioavailability studies where the chemical synthesis of VLC-PUFA PC has also been investigated by using several lipases, including the lipase of *Rhizomucor miehei*. Although VLC-PUFA incorporation into the *sn2* position has been shown, we think the synthetic condition does not mimic the environment of photoreceptor cells. In our manuscript, we focus on the biosynthetic pathway of VLC-PUFAs in phosphatidylcholine (VLC-PUFA PC) in the mouse eye. In fact, the ceramides containing VLC-PUFA were not characterized in this study, and our previous study used an untargeted lipidomics method (<https://www.nature.com/articles/s41587-020-0531-2>).

Regarding agreement with previously reported structures, our findings align with the almost exclusive presence of VLC-PUFAs at the *sn1* position, as explained in the main text. We note that previous studies used phospholipases to analyze positional specificities of VLC-PUFAs, while our approach reveals positional specificity in intact lipids, allowing us to know the combination of acyl chains simultaneously. As such, our study not only confirms previous structural features of VLC-PUFA-containing lipids but also adds more valuable information to consider the biochemical pathway for VLC-PUFA PC. Since the role of ELOVL4 on VLC-PUFA synthesis has already been well established, we focused in our manuscript on the next step, which is their incorporation into glycerophospholipids, as discussed in the response to your first comment.

REVIEWER COMMENTS

Reviewer #1

Thank you for improving the manuscript so far. However, there are still a few questions:

- 1. [major issues 1] In your answer to Reviewer 1's first comment, you mention that the V-shape peak pattern is used as the minimal criterion for detecting the existence of a double bond at a particular position. This would imply that whenever there is indeed a double bond the V-Shape pattern will be observable. Please confirm that this is ALWAYS the case.*

Thank you for your valuable comment. The content we described in our previous point-by-point response letter may have caused some misunderstanding. As you pointed out, the V-shape pattern is by no means an exclusive criterion. Therefore, we have stated in the manuscript as follows:

"For the determination of the C=C positions within complex lipids, the presence of product ions labeled as "C=C high" peak was first confirmed. Then, as unique abundance patterns, such as the V-shape, are not the exclusive criteria, candidates were ranked based on the local correlation coefficient between the experimental and computationally generated in silico spectra related to the acyl chains."

- 2. [minor issues 11] Thank you for revising Fig. 4b (previously 3b). However, I am still missing a legend explaining the meaning of the colors unless they don't really have a meaning. In case of the latter, please add this information to the figure's caption.*

Thank you very much. As you mentioned, the colors in Figure 4b have no specific meaning (they were automatically generated by the Python package). Therefore, we have added the following sentence to the figure's caption:

(b) A sunburst plot summarizing species/tissue-specific lipid database statistics containing collision-cross section (CCS) values. The database comprises lipidomes from 231 biosamples, including humans, mice, plants, and microorganisms. An eye-lipidome table with m/z and CCS values for 525 unique lipids was used to annotate lipids in MSI data analysis. A summary table of peak annotations in the analyzed MSI data is also provided. The abbreviations FA, GL, GP, PR, SP, and ST represent fatty acyls, glycerolipids, glycerophospholipids, prenol lipids, sphingolipids, and sterol lipids, respectively. **The colors in the sunburst plots are automatically generated and do not have any specific meaning.**

- 3. [minor issues 12] In the newly created Table S8 is the B&D referring to the original Bligh and Dyer extraction method or to a modified Bligh and Dyer method? Just want to reconfirm.*

Thank you for your comment. The Bligh and Dyer method we used was a modified Bligh and Dyer method described in our manuscript. We have included this information in the legend of Table S8.

All other previously raised issues that I do not mention here have been adequately addressed by the authors. I still recommend to publish their work on EAD fragmentation annotation after the last few issues have been addressed.

Thank you so much for your constructive comments and suggestions. Our paper was really improved.

Reviewer #2

Takeda et al. have made significant revisions to their manuscript in response to my initial review and, in so doing, have addressed the two substantial issues I raised. The authors are to be commended for their efforts including a large body of additional experimental work that now clarify the advances and residual limitations of their methodology. Based on this I am supportive of publication subject to consideration of the points below which largely relate to providing additional clarity as to the output of the MSDIALv5 analysis of EAD data.

(1) The additional isomer calibration experiments are excellent and are well summarised in Figure 3 (and elsewhere in supplement). This clarifies a number of key points:

a. that the assignment of double bond position is based on changing peak abundance and not unique fragment ions. As such, the spectral fitting is much more reliant on the training set of spectra and thus the confidence in the assignment of the most abundant double bond isomer is likely heavily reliant on a reference spectrum being available. The authors should make clear in their manuscript that the output is only identifying the most abundant double bond isomer and that the confidence in this assignment will diminish for more complex isomeric mixtures and/or for unsaturated lipids for which reference spectra are not available.

Thank you for your insightful comment. We agree that the assignment of double bond positions relies on changes in peak abundance patterns rather than unique fragment ions, making spectral fitting heavily dependent on the availability of reference spectra. To address this limitation, we have added the following sentences to the manuscript.

From these data, we concluded that the current LC-EAD-MS/MS and MS-DIAL 5 technique can accurately annotate the most abundant lipids for co-eluted peaks, as validated for monounsaturated fatty acid-containing PC in this study. However, the assignment of double bond positions is based on changes in peak abundance patterns rather than unique fragment ions, making spectral fitting highly dependent on the availability of reference spectra. Consequently, the confidence in identifying the most abundant double bond isomer diminishes in more complex isomeric mixtures or for unsaturated lipids lacking reference spectra. The accuracy of spectral annotation can also be reduced when isomers co-elute, particularly in the case of C=C positional isomers. This highlights the necessity for further development in sample preparation and chromatography techniques to fully leverage the potential of EAD-MS/MS. Nevertheless, our current data suggests that the abundance of co-eluting lipids can be estimated if synthetic standards are available.

b. the sn-specific marker ions appear to be more exclusive to individual isomers and so are a more sensitive measure of the presence of multiple isomers. The data from biological samples should therefore be annotated to demonstrate the power of this information. That is it is likely that the annotation can reflect where a mixture of isomers is shown to be present without necessarily making a full molecular lipid assignment. That is, it is biologically significant to be able to indicate that both PC 16:0/18:1 and 18:1/16:0 are present without necessarily being able to assign the double bond position to the sn-position.

Thank you for your valuable and encouraging comments. We agree that the sn-specific marker ions are indeed more exclusive to individual isomers and provide a more sensitive measure of the presence of

multiple isomers. To emphasize this strength of our method, we have added the following text to the manuscript.

“Moreover, our results clearly indicate that EAD-MS/MS has an advantage in characterizing sn1/sn2 positional isomers because the sn1-specific fragment ions are less contaminated by charge remote fragment ions related to acyl chains. This allows us to detect the presence of isomeric mixtures, such as both PC 16:0/18:1 and PC 18:1/16:0, without necessarily assigning the double bond position to the sn-position. The ability to annotate such mixtures is biologically significant, providing deeper insight into lipid metabolism and function.”

(2) Line 308 The statement “...lipids with unique structures, for which the presence of isomers is less expected” is not consistent with the data in the current manuscript or the literature. The manuscript shows that isomers are more often present than absent. It is important to note that the methods outlined in this paper are demonstrated to be effective for identification of the most abundant isomer but do NOT exclude the possibility of other isomers being present. Indeed, even the case study of the VLC-PUFAs should be used to illustrate this. Based on the signal to noise achieved in these spectra and the calibration curves now included the authors would do well to point out explicitly that they cannot rule out the presence of other isomers at up to 10%.

We agree that the statement was not accurate and could lead to misunderstandings. Isomerism is indeed common in lipids, and our data demonstrate that isomers are often present. We excluded the following sentence.

“Therefore, the method is particularly robust for the analysis of lipids with unique structures, for which the presence of isomers is less expected.”

In addition, we added the following sentence in the result and discussion section of VLC-PUFA PC.

“Moreover, our results indicated that all the top hit candidates with sn- and C=C-positional information contained n-3-VLC-PUFA at the sn1-position, while our method does not exclude the possibility of other isomers being present.”

(3) Line 66 “electrospray” not “electron spray”

Thanks! We changed it accordingly. Thank you so much for your all constructive comments and suggestions. Our paper was really improved.